# Quantitative live cell imaging reveals influenza virus manipulation of Rab11A transport through reduced dynein association

Amar R. Bhagwat [1], Valerie Le Sage[1], Eric Nturibi[1], Katarzyna Kulej[2], Jennifer Jones [1], Min Guo[3], Eui Tae Kim [2], Benjamin A. Garcia[4,5], Matthew D. Weitzman[2,5,6], Hari Shroff[3] & Seema S. Lakdawala [1,7]*

Assembly of infectious influenza A viruses (IAV) is a complex process involving transport from the nucleus to the plasma membrane. Rab11A-containing recycling endosomes have been identified as a platform for intracellular transport of viral RNA (vRNA). Here, using high spatiotemporal resolution light-sheet microscopy (~1.4 volumes/second, 330 nm isotropic resolution), we quantify Rab11A and vRNA movement in live cells during IAV infection and report that IAV infection decreases speed and increases arrest of Rab11A. Unexpectedly, infection with respiratory syncytial virus alters Rab11A motion in a manner opposite to IAV, suggesting that Rab11A is a common host component that is differentially manipulated by respiratory RNA viruses. Using two-color imaging we demonstrate co-transport of Rab11A and IAV vRNA in infected cells and provide direct evidence that vRNA-associated Rab11A have altered transport. The mechanism of altered Rab11A movement is likely related to a decrease in dynein motors bound to Rab11A vesicles during IAV infection.

[1] Department of Microbiology and Molecular Genetics, University of Pittsburgh School of Medicine, 450 Technology Drive, Pittsburgh, PA 15219, USA. [2] The Children's Hospital of Philadelphia Research Institute, 3501 Civic Center Dr., Philadelphia, PA 19104, USA. [3] Section on High Resolution Optical Imaging, National Institute of Biomedical Imaging and Bioengineering, National Institutes of Health, 13 South Drive, Building 13, Bethesda, MD 20892, USA. [4] Department of Biochemistry and Biophysics, University of Pennsylvania Perelman School of Medicine, 3400 Civic Center Blvd, Building 421, Philadelphia, PA 19104, USA. [5] Epigenetics Institute, University of Pennsylvania Perelman School of Medicine, 3400 Civic Center Blvd, Philadelphia, PA 19104, USA. [6] Department of Pathology and Laboratory Medicine, University of Pennsylvania Perelman School of Medicine, Philadelphia, PA 19104, USA. [7] Center for Vaccine Research, University of Pittsburgh School of Medicine, Pittsburgh, PA 15219, USA. *email: lakdawala@pitt.edu

nfluenza A virus (IAV) is a negative-strand RNA virus with a segmented genome. Replication of each IAV genome segment occurs in the nucleus independently of each other[1,2]. The segmented genome requires assembly of all eight segments prior to release of progeny virions from the plasma membrane of an infected cell[3–5]. Viral RNA is bound by the virally encoded nucleoprotein (NP) and the heterotrimeric viral polymerase, composed of PA, PB1, and PB2, at the 5′ and 3′ ends to form a pan-handle structure. Together, the RNA, NP scaffold, and viral polymerase form one viral ribonucleoprotein complex (vRNP) per segment[6]. We and others have used immunostaining and fluorescence in situ hybridization (FISH) to demonstrate that vRNP segments are sequentially assembled in the cytoplasm on their way to the plasma membrane[7,8].

Intracellular transport of vRNP from the nucleus to the plasma membrane is thought to be mediated by Rab11A, a host-cell small GTPase[9–11], via direct protein–protein interaction with the PB2 subunit of the polymerase[12,13]. Rab11A specifically marks recycling endosomes (RE), which sort and transport cargo slated for release from the apical cell membrane[14,15]. Rab11A recruits various molecular motors to RE through interactions with its corresponding family interacting proteins (Rab11-FIPs)[16]. Rab11-FIPs can associate with both actin and microtubule-associated motor proteins, indicating that Rab11A-RE can use multiple cytoskeletal networks for transport[17–20]. Thus, IAV vRNP segments are thought to transport to the plasma membrane along cytoskeletal filaments via Rab11A-RE, but the details of transport to the viral budding sites are poorly understood.

We and others have shown that depolymerization of the microtubule network fails to significantly abrogate viral replication[13,21,22]. Intriguingly, using FISH and immunostaining in fixed cells, we have observed a marked decrease in diffraction-limited colocalization between Rab11A and vRNP in the absence of an intact microtubule network[22]. These data suggest a reduction in Rab11A–vRNP association and the existence of an alternative, Rab11A and microtubule-independent, mode of transport for vRNP segments.

Robust time-resolved volumetric imaging studies of intracellular dynamics of viral and host proteins have been limited due to competing requirements on imaging speed, photodamage, and spatial resolution using conventional confocal microscopy. Dual-view inverted selective-plane illumination microscopy (diSPIM) has enabled imaging live cells at high temporal resolution (200-Hz frame rate, 2 volumes/s) with isotropic resolution (~330 nm) over long time intervals (>12 h) without imparting the photo-toxicity or bleaching inherent to spinning-disk confocal microscopes[23,24].

Here, we use a diSPIM to quantify and statistically analyze the cellular motion of fluorescently tagged Rab11A and IAV vRNP in live cells. We conclusively demonstrate that Rab11A motion in lung epithelial cells occurs primarily on microtubules. In contrast, IAV vRNP maintains a significant fraction of their motion in the absence of microtubule filaments, confirming the presence of a microtubule-independent mechanism of transport. By quantifying the motion of Rab11A in cells infected with IAV and respiratory syncytial virus (RSV), we demonstrate that these viruses differentially modulate the motion of intracellular Rab11A, suggesting the use of different mechanisms. Based on these data, we predict an important role for Rab11A-RE in the transport and assembly of multiple RNA viruses that can be manipulated to interfere with viral infection. Two-color imaging of Rab11A and IAV vRNP in the same cell reveals co-transport. Comparing the movement of Rab11A foci colocalized with vRNP with foci that are not colocalized confirmed altered transport of Rab11A associated with IAV vRNP. Using immunoprecipitation and mass spectrometry, we demonstrate a decrease in dynein motors associated with Rab11A during IAV infection. Taken together, we propose a model in which Rab11A transport is altered during infection due to a change in the ratio of motor proteins on Rab11A-RE. More generally, we provide a versatile framework and pipeline to quantitatively examine the intracellular transport dynamics of foci with high spatiotemporal resolution.

## Results

### Analyzing three-dimensional movement at single-cell level.
In this report, we track the movement of RE and IAV segments in A549 lung epithelial cells. Successful vesicle tracking requires imaging with high spatiotemporal resolution followed by robust tracking algorithms. Tracking over extended periods allows us to apply statistical analysis to study cellular motion. Figure 1 illustrates the workflow we developed for tracking, by analyzing and distilling track data derived from 3D images of single cells (also see Supplementary Movie 1). Our diSPIM datasets feature isotropic resolution (<350 nm, for details and characterization of the diSPIM see Materials and Methods, Supplementary Fig. 1 and Supplementary Table 1) and are obtained at ~1.4 Hz volumetric temporal resolution to capture fast-moving vesicles such as Rab11A-RE.

Each cell produces thousands of tracks during our imaging experiments and each track is analyzed using multiple motion parameters (described below). The cumulative distribution function of each parameter was used to distill the intracellular movement information into a single number for comparison across cells and treatment groups (Fig. 1c, d). Our imaging analysis pipeline quantifies several parameters characterizing intracellular movement: (1) *track length*: sum of lengths of individual segments that make a continuous track (μm), (2) *track duration*: total number of timeframes for a track, multiplied by time interval between frames (seconds), (3) *track displacement*: minimum distance between track beginning and end points, (4) *track speed*: ratio of track length to track duration (μm/s), (5) *track velocity*: ratio of track displacement to track duration (μm/s), and (6) *normalized standard deviation of speed*: measures how speed varies over a given track and is calculated as a ratio of standard deviation of speed within a track to average track speed (unitless).

To quantitatively distinguish between local and long-range transport, we calculated two additional parameters: (1) *track straightness*: ratio of displacement to track length, ranges from zero (highly convoluted motion) to one (highly directional motion) and (2) *arrest coefficient*: fraction of steps composing the track that are smaller than 300 nm—the diffraction limit—effectively identifying stalled motion. Arrest coefficient ranges from zero (unimpeded motion/no stalling) to one (completely stalled motion). Rab11A puncta movement is very dynamic with multiple fission or fusion events and intervening stalled or tethered states (Fig. 2, Supplementary Movie 2). On multiple occasions, the Rab11A foci merge into or split from already-stalled foci. However, our tracking algorithm is unable to score fission or fusion events, and a new track is created or ended respectively after each of these events. Therefore, a disproportionate increase in the number of tracks without a corresponding increase in the total number of spots (or increased arrest coefficient) may be indicators of increased fusion and fission events. Using these methods, we effectively quantified the impact of various treatments and virus infection on cellular transport of Rab11A and influenza vRNP.

### Rab11A moves predominantly on microtubules in A549 cells.
Previous imaging studies of Rab11A-RE in fixed cells treated with

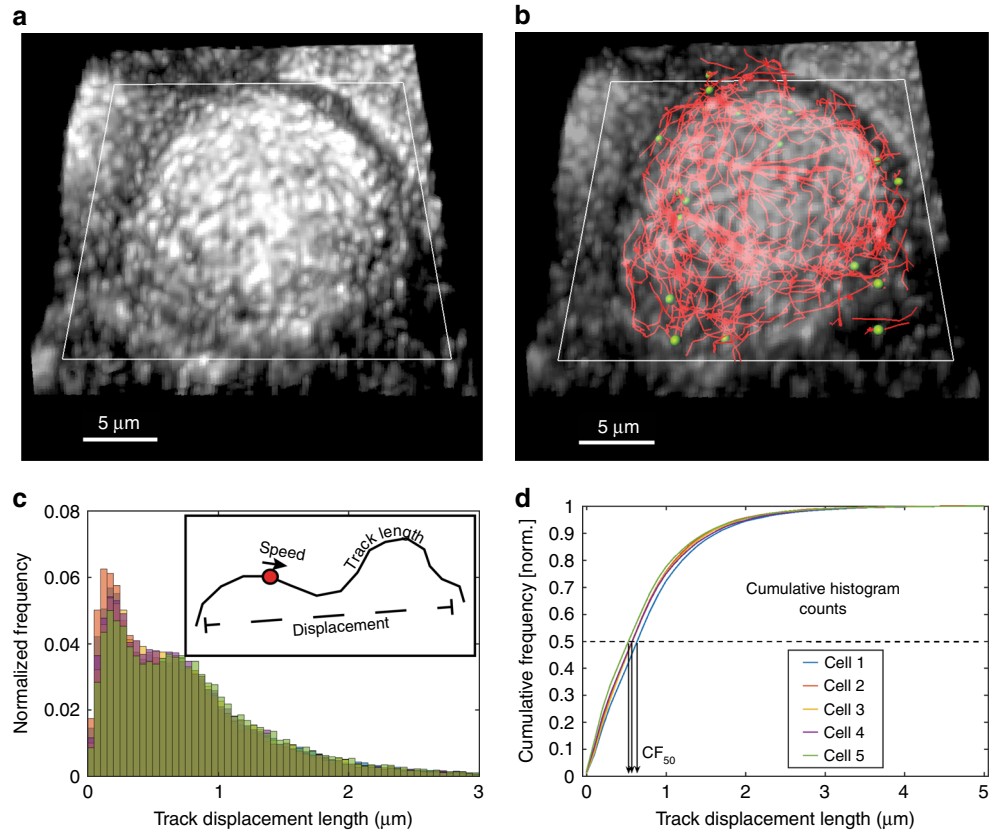

**Fig. 1 Workflow to analyze intracellular motion in three dimensions (3D). a** Dual-view volumetric images are fused and deconvolved to obtain a single volumetric image with ~330-nm isotropic resolution[23]. Here, a diSPIM maximum intensity projection is shown for a WSN-infected A549 cell stably expressing GFP-tagged Rab11A (GFP-Rab11A). The coverslip plane (white rectangle) is also indicated. **b** Fluorescent puncta are identified and tracked over time in Imaris. For clarity, only tracks with displacement >2 μm are shown (red lines). Green spheres show GFP-Rab11A spot centers from a single time frame corresponding only to the shown tracks (for clarity, not all identified foci are shown here). **c** Distributions of motion parameters are binned into histograms, one histogram per cell parameter (e.g., track displacement histograms from five cells are shown here as a representative example). The inset illustrates parameters such as speed, track length, and displacement. All motion parameters are defined in the text. **d** Motion parameters (a representative example for track displacement is shown here) are compared across treatments by generating the corresponding cumulative histograms and measuring $CF_{50}$—the x-axis value at which the cumulative frequency is 50%. This value summarizes the given motion parameter for that cell. One-way ANOVA with Tukey's multiple tests is used to perform comparisons on $CF_{50}$ values to gauge the effect of a treatment.

nocodazole, a microtubule-depolymerizing drug, have observed a mislocalization of Rab11A-positive puncta from the apical surfaces of cells to the basal periphery of cells[13,22,25]. To examine the dynamics of Rab11A in human lung cells, we performed live imaging of A549 cells stably expressing Rab11A protein tagged with GFP on the N terminus (henceforth referred to as "A549 GFP-Rab11A"). Cells were treated with either a microtubule-depolymerizing drug (nocodazole) or actin-depolymerizing drug (latrunculin A) or the drug vehicle (dimethyl sulfoxide, DMSO) as a mock control.

Cellular transport of Alexa-568-tagged transferrin in GFP-Rab11A cells was found to be similar to unmodified A549 cells by most measures except mean speeds, which showed a small, albeit, statistically significant difference (Supplementary Fig. 2). Thus, in this report, we do not directly compare the results obtained in unmodified A549 cells with those obtained in GFP-Rab11A-expressing A549 cells.

Live-cell imaging of DMSO-treated A549 GFP-Rab11A cells (Supplementary Movie 3) illustrates apical concentration of GFP-Rab11A and dynamic movement of puncta within a cell. In comparison, nocodazole treatment largely abrogated long-distance motion of Rab11A puncta (Supplementary Movie 4), disrupted the microtubule network in A549 GFP-Rab11A cells, and caused mislocalization of GFP signal from the apical surface

to the cellular periphery (Supplementary Fig. 3). Comparison of GFP-Rab11A movement between DMSO- and nocodazole-treated cells showed significantly decreased mean track speed, velocity, and straightness of Rab11A puncta (Fig. 3a–c), suggesting a strong reduction in long-distance transport. This observation was corroborated by a reduction in track length (Supplementary Fig. 4b) and displacement (Supplementary Fig. 4c). Finally, arrest coefficient was significantly increased with nocodazole treatment (Fig. 3d), highlighting stalled motion of Rab11A in the absence of intact microtubule filaments. The number of spots and tracks was similar between DMSO- and nocodazole-treated cells (Supplementary Fig. 4a, d), demonstrating that these differences in motion are not due to a decrease in GFP-Rab11A sampling.

To determine the role of actin cytoskeleton in Rab11A movement, we treated A549 cells with latrunculin A to depolymerize the actin network, which was confirmed by staining for phalloidin (Supplementary Fig. 3). Latrunculin A treatment causes severe changes in cellular morphology (Supplementary Fig. 3 and Supplementary Movie 5), possibly due to a reduction in stress fiber-associated focal adhesions responsible for attaching adherent cells to a surface[26]. Depolymerizing the actin network resulted in increased GFP-Rab11A speed and velocity compared with DMSO-treated cells (Fig. 3a, b). In addition, we observed a

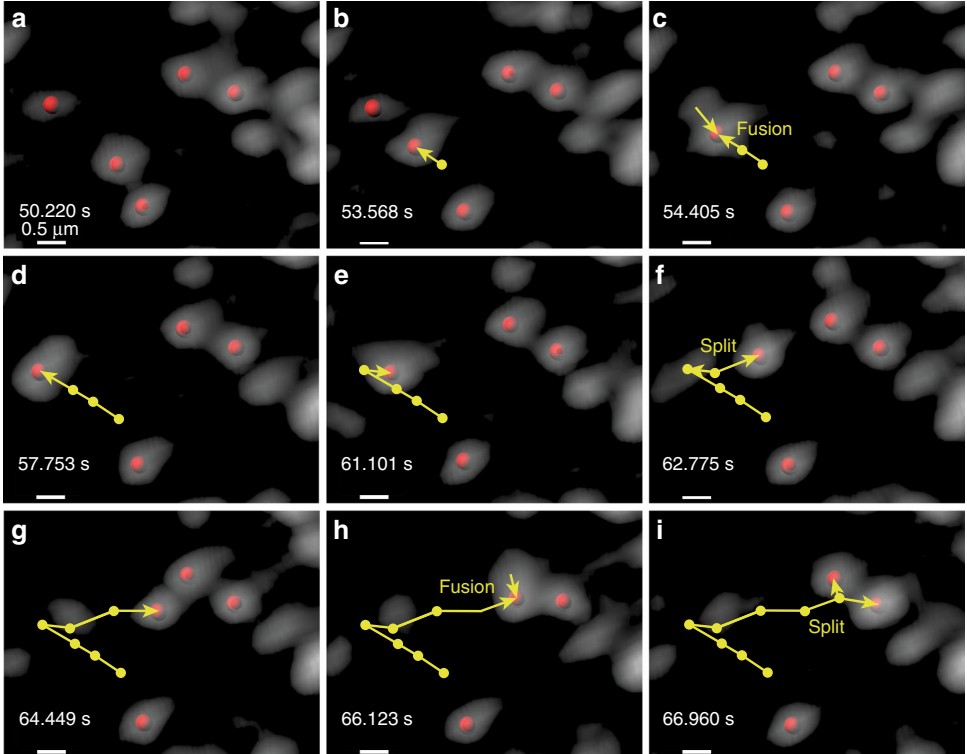

**Fig. 2 Intracellular colocalization and splitting events for GFP-Rab11A foci.** Multiple instances of merging and splitting GFP-Rab11A foci (shown here in gray) can be identified in live-imaging datasets. **a–i** Chronologically ordered time snapshots follow a moving GFP-Rab11A spot merging with and splitting from other GFP-Rab11A foci. The yellow track shows location history and current direction of motion. For clarity, only centers of those GFP-Rab11A foci that take part in this interaction are denoted by red spots. The surrounding vesicles are either stalled or very slow moving. Timestamps show time elapsed since the start of the dataset. Scale bars are 0.5 μm as shown.

significant increase in GFP-Rab11A track straightness (Fig. 3c) and a reduction in the arrest coefficient (Fig. 3d) during latrunculin A treatment. These data (especially increased speed) are consistent with increased microtubule-based movement of vesicles using kinesins and dyneins in the absence of the actin cytoskeletal network[27]. Taken together, our observations imply that Rab11A transports primarily along microtubules within human lung epithelial cells.

**Microtubules not essential for influenza vRNP movement.** Rab11A-containing RE is thought to mediate intracellular transport of IAV vRNP[9–13]. Although depolymerization of microtubules in A549 cells abrogates Rab11A transport (Fig. 3), treatment with nocodazole does not significantly alter replication kinetics of seasonal IAV in cells[13,21,22]. To examine this disparity, we performed live imaging of vRNP movement in nocodazole- and latrunculin A-treated A549 cells to assess the importance of microtubule and actin, respectively. To track IAV vRNP movement, we used a recombinant H1N1 pandemic strain A/California/07/2009 (H1N1pdm) expressing a GFP-tagged PA protein, henceforth referred to as "H1N1pdm PA::GFP". PA is a component of the virally encoded polymerase, and in H1N1pdm PA::GFP-infected cells, the GFP signal colocalize strongly with both NP and vRNA staining (Supplementary Fig. 5a). Thus, examining the movement of GFP-tagged PA protein (henceforth referred to as "PA::GFP") can be used as a surrogate for vRNP intracellular dynamics.

We confirmed that treatment with nocodazole and latrunculin A did not impact the replication kinetics of H1N1pdm PA::GFP virus in A549 cells (Supplementary Fig. 5b). Therefore, this system accurately reflects our previous observations with wild-type H1N1pdm viruses[22]. The impact of drug treatment on cytoskeletal filaments was confirmed by immunofluorescence for actin or microtubules (Supplementary Fig. 6). As we have previously observed[7], imaging and tracking of PA::GFP puncta in DMSO-treated cells show robust vRNP movement with multiple colocalization and splitting events throughout the cytoplasm (Supplementary Fig. 7 and Supplementary Movie 6). Remarkably, depolymerization of microtubules has little impact on vRNP movement, since significant movement can be observed in live imaging of PA::GFP in nocodazole-treated cells (Supplementary Movie 7). This observation is in stark contrast to the movement of GFP-Rab11A in nocodazole-treated cells (Fig. 3 and Supplementary Movie 4). Quantification of the PA::GFP motion does reveal a statistically significant reduction in the average speed, velocity, and straightness of PA::GFP puncta in the absence of intact microtubule filaments (Fig. 4a–c). Previous studies have also found that treatment of infected cells with nocodazole decreased long-range motion of influenza viral RNA[28]. However, when compared with the effect on Rab11A movement (Fig. 3), nocodazole treatment has a much smaller impact on the movement of PA::GFP and a significant portion of the speed and velocity is retained. Finally, no difference in arrest coefficient was observed between DMSO and nocodazole treatments (Fig. 4d), which implies that PA::GFP puncta do not stall more in the absence of microtubules. No significant differences were observed between nocodazole treatment and DMSO for any other PA::GFP movement parameters (Supplementary Fig. 8).

In contrast, when H1N1pdm PA::GFP-infected A549 cells are treated with latrunculin A (Supplementary Movie 8), the dynamics of PA::GFP puncta mirror those of the GFP-Rab11A under latrunculin A treatment (Fig. 3). The average speed and velocities of PA::GFP puncta in latrunculin-treated cells show a

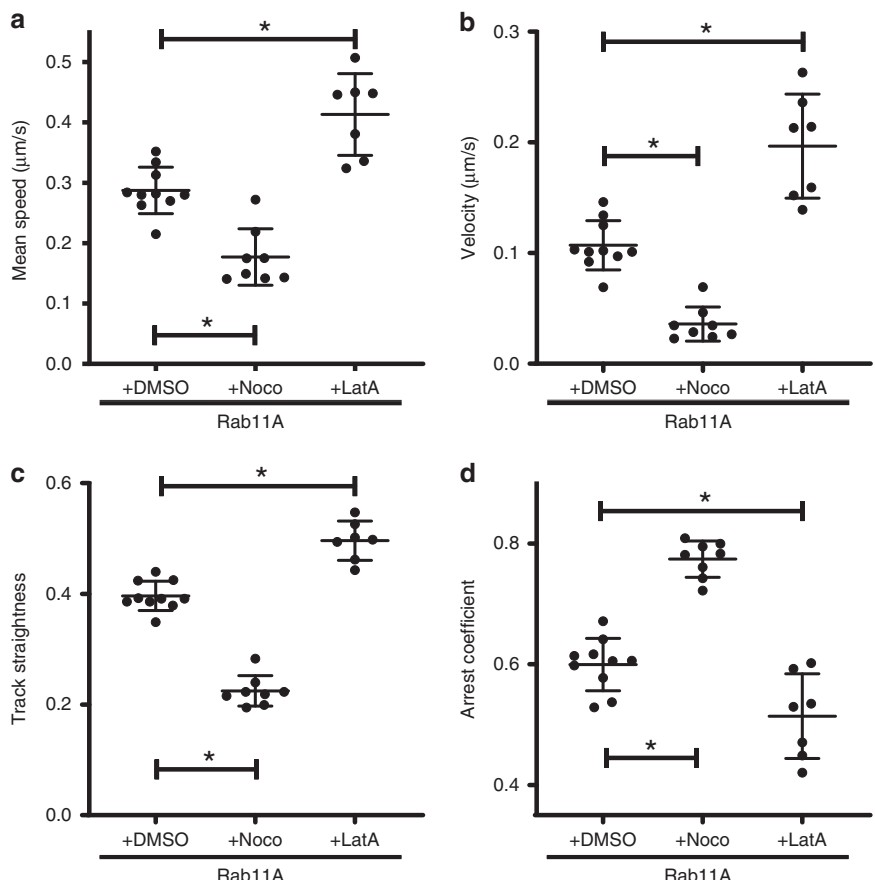

**Fig. 3 Motion dynamics of Rab11A under the effect of cytoskeletal depolymerizing drugs.** A549 stable cells expressing GFP-Rab11A were treated with nocodazole (16 µM), latrunculin A (1 µM), or DMSO (control) and imaged 4 h post treatment. Imaging is performed at 37 °C on a temperature-controlled stage. Each datapoint represents an individual cell (DMSO: $n = 10$ cells, nocodazole: $n = 8$ cells, and latrunculin A: $n = 7$ cells) using $CF_{50}$ for the corresponding motion parameters: **a** track mean speed, **b** track mean velocity, **c** track straightness, and **d** track arrest coefficient. Image processing and analysis were performed as described in the "Methods" section. Mean ± standard deviation (SD) is shown per group (* implies $p < 0.05$, comparisons performed using one-way ANOVA with Tukey's multiple tests). Source data are provided as a Source Data file.

statistically significant increase compared with DMSO control (Fig. 4a and b). This increase is likely due to an increase in track displacement and track length (Supplementary Fig. 8b, c) and a decrease in track durations (Supplementary Fig. 8e). Track straightness also shows a slight increase (Fig. 4c), while the arrest coefficient shows a small decrease (Fig. 4d). These data demonstrate that IAV vRNP retains movement in cells lacking an intact actin network. Analyzing the locations of Rab11A and IAV vRNP foci in fixed cells reveals a decrease in their colocalization in the absence of microtubules (nocodazole treatment) when compared with DMSO controls (Supplementary Fig. 5c). Intriguingly, we also observed increased colocalization in the absence of actin filaments (latrunculin A treatment). Thus, in the absence of actin filaments, vRNP strongly associates with Rab11A-RE, which we have shown to transport predominantly on microtubules. However, the presence of robust PA::GFP movement in nocodazole-treated cells confirms the presence of a microtubule-independent mechanism of transport as we and others have previously suggested[22,28].

**Alteration of Rab11A intracellular movement by IAV and RSV.** Many viruses are known to manipulate the cellular transport machinery[29–31]. To study the effect of IAV infection on Rab11A dynamics, we infected A549 GFP-Rab11A cells with A/WSN/33 (H1N1) influenza virus at a high multiplicity of

infection (MOI) and confirmed that close to 100% of the cells were infected during imaging (at 8 h post infection [hpi]) by fixing and staining the imaged cells with an anti-NP antibody (Supplementary Fig. 9). Uninfected A549 GFP-Rab11A cells were imaged as a control. We observed that in the presence of WSN IAV infection, the mean speed, velocity, and track straightness of Rab11A puncta was significantly reduced (Fig. 5a–c), while the arrest coefficient of Rab11A puncta was significantly increased (Fig. 5d), indicating more stalled motion. The reduction in the velocity can be attributed to a significant decrease in the displacement and track lengths of Rab11A-RE when infected by WSN IAV (Supplementary Fig. 10b, c). Our analysis indicates that during IAV infection, GFP-Rab11A not only travels a shorter distance, but also spends more time making that transit (Supplementary Fig. 10e). Therefore, we conclude that IAV slows down Rab11A movement and promotes more stalled motion.

To examine whether alteration of Rab11A transport in A549 cells is unique to IAV, we investigated whether infection with RSV also altered Rab11A movement (for sample dataset, see Supplementary Movie 9). Similar to influenza viruses, RSV is a respiratory pathogen with a negative-sense RNA genome that is capable of replicating in A549 cells[32]. However, while influenza viruses have a segmented genome and replicate in the nucleus, RSV is single-stranded and replicates in the cytoplasm. Rab11A apical-RE-based transport is needed for completion of RSV replication life cycle[33–35]. Thus, examining the dynamics of

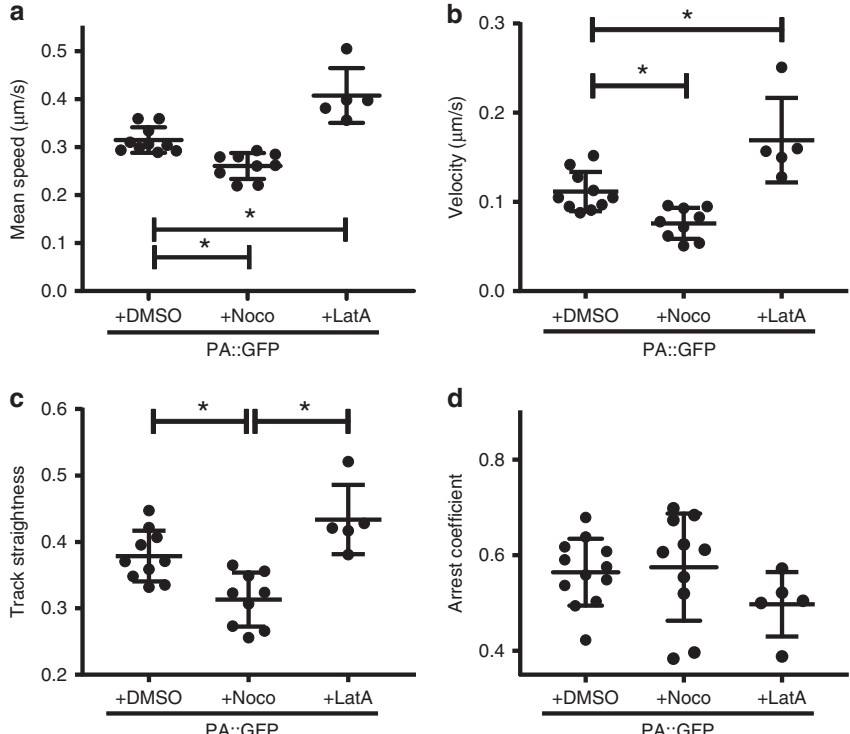

**Fig. 4 Effect of cytoskeleton depolymerization on motion dynamics of influenza vRNP in A549 cells.** Motion parameters of influenza vRNP segments with a fluorescently tagged polymerase. A549 cells were infected with H1N1pdm PA::GFP at multiplicity of infection (MOI) = 1 and treated with drugs 4 h post infection (hpi) to allow unhindered viral entry. Imaging was performed 16 hpi at 37 °C on a temperature-controlled stage. Each datapoint represents an individual cell (DMSO: $n = 10$ cells, nocodazole: $n = 9$ cells, and latrunculin A: $n = 5$ cells) using $CF_{50}$ for the corresponding motion parameters: **a** track mean speed, **b** track mean velocity, **c** track straightness, and **d** track arrest coefficient. Image processing and analysis were performed as described in Methods. Mean ± SD are shown per group (* implies $p < 0.05$, comparisons performed using one-way ANOVA with Tukey's multiple tests). Source data are provided as a Source Data file.

Rab11A during RSV infection will help resolve whether alteration of Rab11A dynamics by IAV is common among other RNA viruses. Surprisingly, the changes in Rab11A motion induced by RSV infection were opposite to those caused by IAV infection. RSV infection caused Rab11A mean speeds (Fig. 5a), velocities (Fig. 5b), and track straightness (Fig. 5c) to increase compared with uninfected cells. However, arrest coefficient was significantly reduced (Fig. 5d). Combined with the observation that track lengths (Supplementary Fig. 10b) and displacements (Supplementary Fig. 10c) are increased, and track durations are shorter (Supplementary Fig. 10e), these observations suggest that RSV alters Rab11A motion so that it is more direct and suffers less obstruction. These data imply that differences in genomic segmentation or perhaps replication location may contribute to differential manipulation of Rab11A dynamics.

To support our hypothesis that segmented viruses manipulate the movement of Rab11A differently than single-stranded RNA viruses like RSV, we examined the movement during infection with influenza B virus (IBV). IBV co-circulates with IAV but is only known to infect humans and possibly seals[36], while IAV has a much larger host range[37]. Like IAV, IBV is also thought to use the Rab11A pathway. We infected A549 GFP-Rab11A cells with IBV at a high MOI and confirmed that close to 100% of the cells were infected during imaging (at 8 hpi) by fixing and staining the imaged cells with an anti-NP antibody (Supplementary Fig. 9). The effect on Rab11A motion when cells are infected with IBV was quantitatively similar to IAV-induced changes (Supplementary Fig. 10, Supplementary Movie 10). This result is intuitive because of the close similarities between IAV and IBV in terms of vRNP segment structure and protein functions. It is nevertheless

an important result because it suggests a universal Rab11A-mediated transport pathway for influenza viral infections. In addition, our data suggest that the Rab11A-RE-based transport is a common pathway that is differentially regulated during distinct respiratory viral infections.

**Two-color imaging demonstrates Rab11A and vRNP co-transport.** In order to examine the co-transport between IAV vRNP and cellular proteins such as Rab11A, we rescued an A/WSN/33 H1N1 virus that encodes for an mRuby-tagged viral PA protein (henceforth called PA::mRuby). We used this virus to infect GFP-Rab11A cells and perform two-color volumetric, live imaging of IAV and Rab11A transport within the same cell using the diSPIM. Figure 6a–c portrays a cross section through the volumetric dataset at a single timepoint as part of the complete dataset represented in Supplementary Movie 11. Figure 6a shows the structure and distribution of GFP-Rab11A-RE moving within a single cell, while Fig. 6b, c shows the signal from IAV vRNP inside the same cell and the corresponding overlap between the two signals respectively. Analysis of colocalized cytoplasmic GFP-Rab11A and PA::mRuby foci within the same cell revealed similar track trajectories from GFP-Rab11A (green spheres) and PA::mRuby (magenta spheres) (Fig. 6d, green and magenta lines). A prior study used transfection of tagged NP and Rab11A with a vRNA segment in 293T cells to show co-transport of Rab11A and vRNP[13]. Here, we analyze co-transport between Rab11A and IAV vRNP during a productive infection, in a relevant cell line, and with three-dimensional resolution.

While a large number of PA::mRuby puncta colocalize with Rab11A spots, greater than 50% of the total PA::mRuby spots do

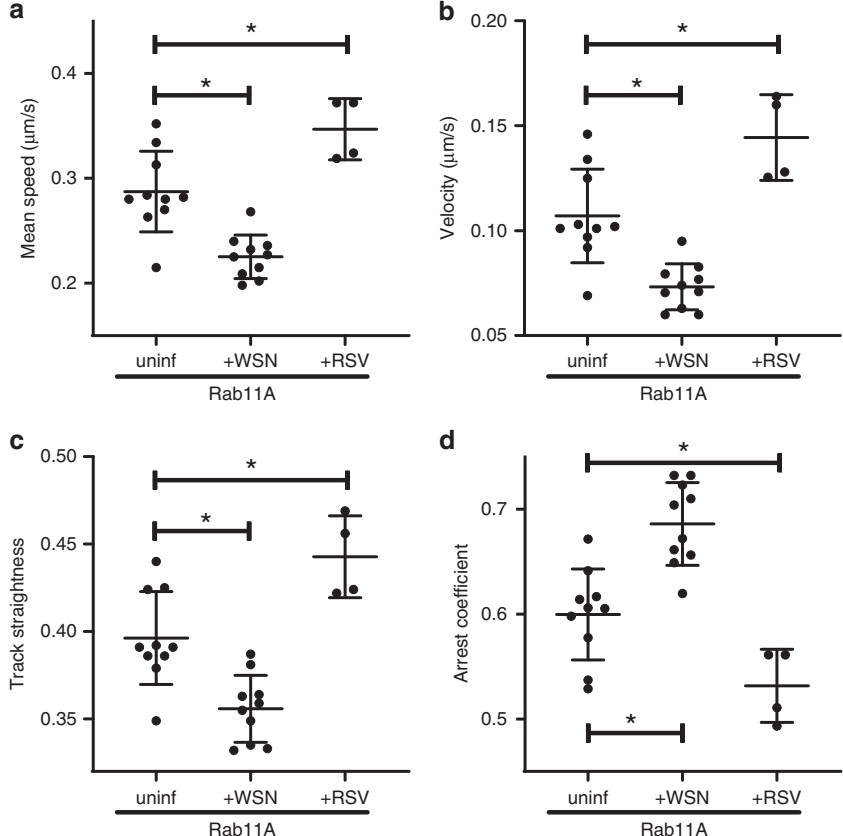

**Fig. 5 Effect of viral infection on Rab11A motion: comparing RSV infection with IAV (WSN) infection.** A549 cells stably expressing GFP-tagged Rab11A were infected with lab-adapted H1N1 strain (A/WSN/33) at MOI = 3 to allow a majority of cells to be infected. Imaging was performed 8 hpi at 37 °C on a temperature-controlled stage. Similarly, A549 cells stably expressing GFP-tagged Rab11A were infected with recombinant RFP-expressing RSV[53] at 0.1 plaque-forming units (pfu)/cell and imaging performed at 16 hpi. Infection with specific strains is noted on the x-axis along with uninfected control (uninf.). Each datapoint represents an individual cell (uninfected: n = 10 cells, WSN: n = 10 cells, and RSV: n = 4 cells) using $CF_{50}$ for the corresponding motion parameters: **a** track mean speed, **b** track mean velocity, **c** track straightness, and d, track arrest coefficient. Image processing and analysis were performed as described in Methods. Mean ± SD are shown per group (* implies $p < 0.05$, comparisons performed using one-way ANOVA with Tukey's multiple tests). Source data are provided as a Source Data file.

not colocalize with GFP-Rab11A spots (Supplementary Fig. 11f) and vice versa. Therefore, isolating the effect of infection on Rab11A motion could be confounded by analysis of all of the GFP-Rab11A puncta within a cell due to this nonhomogeneous association with IAV vRNP. Two-color imaging of GFP-Rab11A and vRNP provides an important advantage by teasing apart the specific impact of IAV infection on vesicular transport within a single infected cell. We separated the intracellular foci into three categories: (1) GFP-Rab11A alone, (2) colocalized GFP-Rab11A and PA::mRuby, and (3) PA::mRuby alone. We independently tracked all three of these populations and analyzed their tracks via our pipeline (see Methods). Our preliminary analysis shows a statistically significant reduction in the speed and displacement of GFP-Rab11A spots that colocalize with PA::mRuby spots, as compared with both single Rab11A and PA::mRuby spots (Fig. 6e, f). In addition, we also observed that track lengths and track durations for PA::mRuby spots were significantly lower than those for Rab11A singletons (Supplementary Fig. 11a, b). Taken together, these data conclusively demonstrate that Rab11A associated with vRNP moves slower than Rab11A foci not associated with vRNP, as also suggested by our single-color imaging datasets (Fig. 5). Differences in PA::mRuby movement between foci colocalized with Rab11A and alone were also observed, as PA::mRuby-alone foci had a higher displacement and moved at a faster speed (Fig. 6e, f), indicating a potential

Rab11A-independent mode of transport. However, these differences are difficult to extrapolate, since endogenous Rab11A is still present in these cells and could be binding to PA::mRuby not colocalized with GFP-Rab11A.

The transport parameters obtained from two-color datasets cannot be directly compared with those from the single-color analysis presented earlier—there is a considerably longer time interval between acquisition of two-color cell volumes (~2.8 s) compared with single-color datasets (~0.7 s/volume, see Methods). This makes tracking and transport analysis challenging and may distort many of the track values. Further work using faster time resolution is necessary to decipher the nature of interaction between Rab11A and vRNP segments.

**IAV infection reduces the association of dynein on Rab11A.** The regulation of vesicular movement may provide clues as to how viruses could differentially manipulate cellular transport. Rab11A specifically associates with different motor proteins via the corresponding FIPs[17,19,38–40]. The two microtubule motors: kinesin (plus (+) end directed) and dynein (minus (−) end directed) have vastly different capabilities, and endosomes traveling on microtubules are thought to carry approximately one kinesin per 4–6 dynein motors[41–43]. These asymmetric teams of multiple opposing motors, through an unknown mechanism, coordinate long-distance travel involving frequent direction

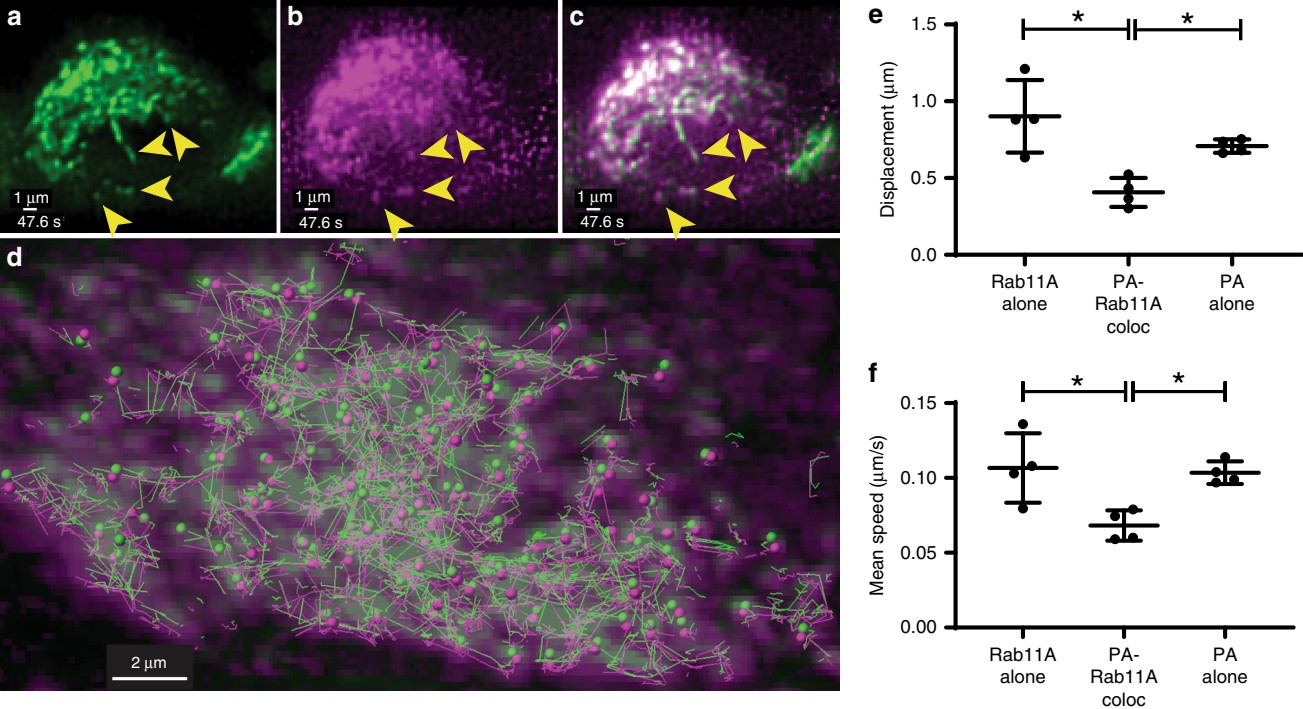

**Fig. 6 Two-color imaging of PA::mRuby-infected GFP-Rab11A cells.** A549 cells stably expressing GFP-tagged Rab11A were infected with recombinant mRuby-tagged PA expressing IAV A/WSN/33. Cross-section through the volume depicting the signal from **a** GFP-Rab11A, **b** mRuby::PA, and **c** merged channels. Note the absence of GFP-Rab11A signal from the nuclear volume, whereas PA::mRuby comprises a homogeneous signal from within the nucleus. Yellow arrowheads point out the corresponding colocalizing puncta in the three channels corresponding to co-transporting species (see Supplementary Movie 11). **d** Overlap between maximum projection of GFP-Rab11A (green) and PA::mRuby (magenta) channels at a single timepoint (0 s) for the same dataset. Colocalizing puncta from each species at that timepoint are depicted by spheres of corresponding colors, while lines of corresponding colors depict their tracks over all times. Significant co-travel between colocalizing puncta is observed on account of close proximity between individual track segments traveling along the same direction. In the right panels, each datapoint represents an individual cell ($n = 4$ cells) using $CF_{50}$ for the corresponding motion parameters: **e** track displacement and f, track mean speed for Rab11A singletons, Rab11A colocalized with PA::mRuby and PA::mRuby singletons. Mean ± SD are shown per group (* implies $p < 0.05$, comparisons performed using one-way ANOVA with Tukey's multiple tests). Source data are provided as a Source Data file.

reversals, stalling, splitting and fusion of endosomes, and detachment from microtubules[42–45]. During live-cell imaging studies, we have observed GFP-Rab11A puncta splitting or merging events in close association with stalling events (Fig. 2 and Supplementary Movie 2). Stalling and splitting behavior could result in a higher arrest coefficient as seen with IAV and IBV infection. Therefore, motor stoichiometry may be a target of viral manipulation.

Accordingly, we infected GFP-Rab11A A549 cells with either H1N1pdm (A/CA/07/2009) virus or mock infected, and at 16 hpi the GFP-Rab11A protein was immunoprecipitated from cell lysates with anti-GFP antibody. Mass spectrometry analysis of two independent replicates revealed a wide range of host proteins with diverse functions, from RNA biogenesis to cellular vesicle transport, to be associated with Rab11A in mock- and IAV-infected cells (Fig. 7a). Gene Ontology (GO) analysis of all Rab11A-interacting proteins identified by mass spectrometry are detailed in Fig. 7a. Many of the RNA biogenesis proteins that we identified reportedly associate with influenza virus PB2 and are packaged into influenza virions[46,47]. However, in this study we were specifically interested in vesicle transport proteins, and considered for validation proteins in this category with $p$ value < 0.05 and $\log_2$-fold decrease of ≤−1 (minimum twofold decrease). Interestingly, we found a significant reduction in the association with two members of the dynein family (DYNC1I2 and DYNC1L12), FIP2 and myosin motor (MYO1D) (Fig. 7b). No other changes were observed for Rab11A FIP 1 or 5 proteins,

kinesin proteins (KIFs), or myosin proteins, even though they were detected in the pulldown (Supplementary Data 1). Surprisingly, FIP 3 and 4 proteins were not detected in the input or in Rab11A pulldown (Supplementary Data 1). Examination of available proteomic databases revealed that these proteins are largely undetectable in lung cells[48–51]. Low abundance of FIP 3 and 4 in lung cells would prevent detection by our mass spectrometry procedure, and can explain the absence of these proteins in our dataset. Analysis of immunoprecipitated input by mass spectrometry confirms that the total levels of the dynein proteins are not altered during infection (Fig. 7c), suggesting that decreases in Rab11A and dynein association are specific to viral infection.

To validate that Rab11A had decreased association with dynein during IAV infection, we performed western blot analysis for dynein levels on additional GFP-Rab11 immunoprecipitation samples (Fig. 7d). We observed a large reduction in the amount of dynein associated with Rab11A in IAV-infected cells. A small, but consistent, decrease in GFP-Rab11A levels was observed during H1N1pdm infection, confirming a similar finding from our mass spectrometry results (Supplementary Data 1). However, we demonstrate association with viral PB2 protein as expected (Fig. 7b). These data confirm that IAV infection decreases association of dynein with Rab11A, thus altering the motor stoichiometry of Rab11A-RE during IAV infection.

Recent work using mitochondrial targeting of FIP proteins suggested that IAV vRNP segments outcompete FIP2 for binding

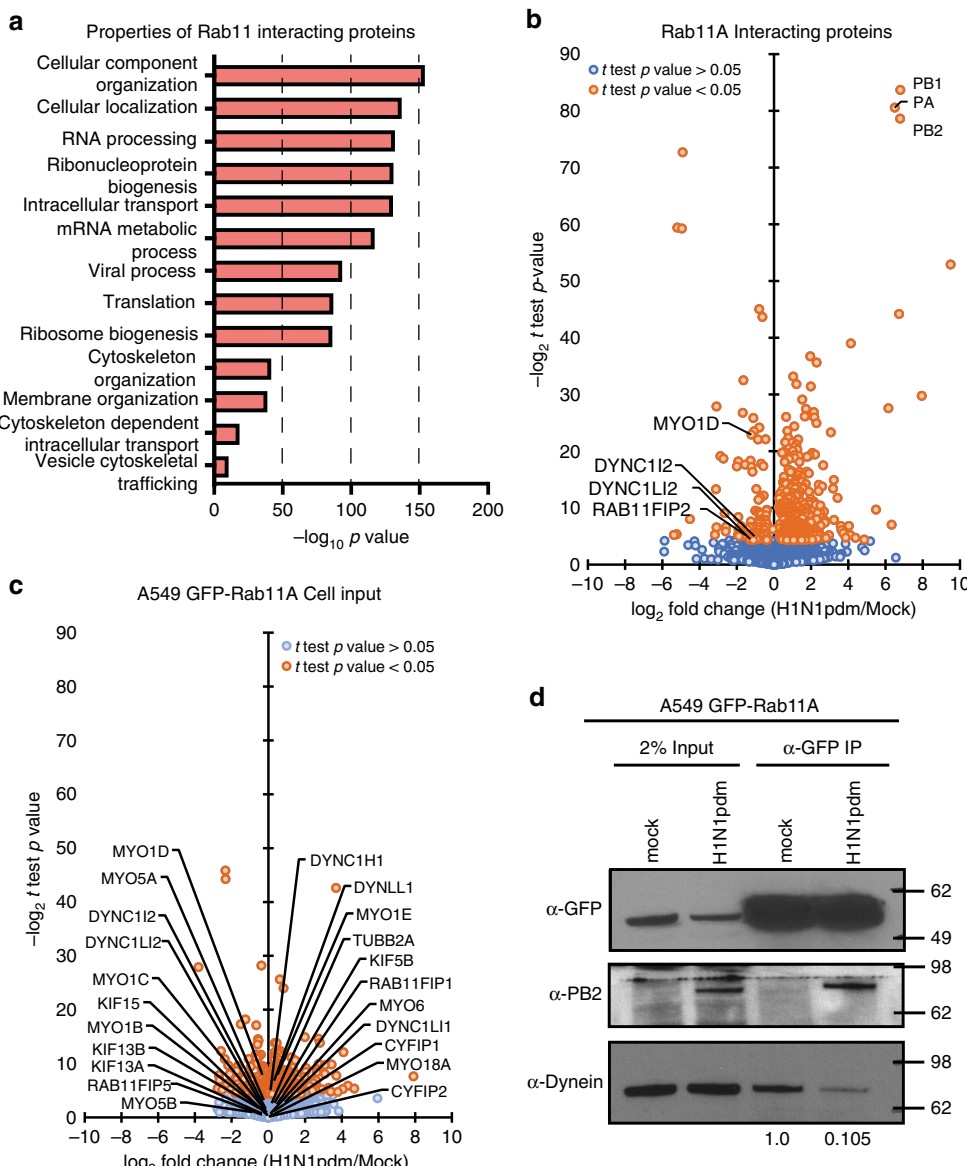

**Fig. 7 Rab11A association with dynein motors is decreased during IAV infection.** Mass spectrometry of immunoprecipitation (IP) input or α-GFP IP from A549 GFP-Rab11A cells either uninfected or infected with H1N1pdm. **a** Gene Ontology analysis of Rab11A-interacting cellular proteins revealed by immunoprecipitation–mass spectrometry. **b** Change in Rab11A-interacting proteins as a ratio of infected to mock cells. Vesicle transport-related proteins significantly altered during infection and viral polymerase proteins are highlighted. **c** Ratio of host proteins in IP input as a ratio of infected to mock cells. **d** Validation and quantification of dynein levels bound to Rab11A in mock or H1N1pdm-infected cells. Relative band intensity of dynein is noted below the western blot and molecular weight size markers indicated to the right. Source data are provided as a Source Data file.

with Rab11A[52]. Consistent with this finding, we observed a significant decrease in the association with FIP2 binding to Rab11A during IAV infection (Fig. 7b). However, the levels of other detectable FIPs were not altered. FIP2 is thought to mediate association with myosin 5[20], which was not altered in our pull-down analysis (Supplementary Data 1). Interestingly, it has also been reported that Rab11A-RE bound to IAV vRNP uses KIF13A to travel to budding regions on the cell plasma membrane[53]. However, in agreement with Ramos-Nascimento et al. who were unable to observe direct association of KIF13A with Rab11A, we did not detect KIF13A in our Rab11A pull-down mass spectrometry analysis. Additional immunoprecipitation studies did not detect any KIF13A associated with GFP-Rab11A, even though KIF13A was detected in the cytosolic input samples (Supplementary Fig. 12). Previous data implicating KIF13A in Rab11A transport were derived mainly from the percentage of NP signal

found within 2 μm from the cell periphery of static images from a confocal slice[53]. However, our results reveal a robust disruption in the amounts of dynein associated with Rab11A during IAV infection, suggesting that IAV infection alters the stoichiometry of Rab11A-RE-associated motors by reducing association of minus-end directed dynein motors.

## Discussion

In this paper, we provide a framework for three-dimensional mapping and analysis of fluorescently tagged cellular and viral components in live cells using high spatiotemporal resolution, low phototoxicity light-sheet imaging. Specifically, we examined the role of cytoskeletal proteins in intracellular transport of Rab11A and influenza vRNP. We observed a clear preference for Rab11A movement along microtubules that was not conserved

**Fig. 8 Schematic of altered Rab11A movement during influenza viral infection.** Based on the data presented in this study, we propose that influenza vRNP will displace dynein motors on Rab11A-containing vesicles to alter movement along microtubules.

for IAV vRNP transport. Subsequent experiments revealed differential viral manipulation of Rab11A-RE cellular transport properties during infection by different negative-stranded RNA viruses. Taken together, these data demonstrate that Rab11A movement is altered during viral infection by distinct RNA viruses, but that IAV can also utilize a microtubule-independent mechanism of vRNP intracellular transport. Our study highlights the redundancy in viral RNA intracellular transport mechanisms and could aid development of pan-viral therapeutics.

During RSV infection, Rab11A motion was highly directional (straightness close to 1), which would benefit a single-stranded genomic virus such as RSV. On the other hand, infection with multisegmented RNA viruses such as IAV and IBV resulted in increased stalling of Rab11A motion, indicated by an increase in arrest coefficients and a decrease in mean speed (Fig. 5, Supplementary Fig. 10). Stalled Rab11A movement in influenza virus-infected cells may be conducive to an increase in vesicular fission and fusion events that facilitates vRNP segment–segment interactions, thus enhancing assembly of all eight heterologous vRNP segments into a supramolecular structure for packaging[3]. Further studies examining Rab11A dynamics during infection with other RNA viruses that use Rab11A transport machinery, such as Sendai virus[54,55], will help further elucidate the conserved role of Rab11A during viral infection.

We quantified Rab11A-interacting proteins using immunoprecipitation and mass spectrometry techniques, and identified a reduction in Rab11A–dynein association as the mechanism behind the altered Rab11A movement during IAV infection. Rab11A specifically associates with motor proteins through their respective adaptor FIP proteins[17,19,38–40]. Recent work using mitochondrial targeting of FIP proteins found that IAV vRNP segments may outcompete certain FIPs for binding with Rab11A[52], which could lead to a change in the distribution of motors on the Rab11A-containing RE. In this study, we could not determine whether a reduction in dynein was a result of reduced Rab11A association with the corresponding dynein FIP, Rab11-FIP 3, as the latter is not detectable in lung cells based on data in the Human Protein Atlas. An alternative hypothesis is that FIP2, which has decreased Rab11A association during IAV infection in our mass spectrometry studies, functions as a dynein adaptor for Rab11A in A549 cells similar to reports in MDCK cells[56]. In this scenario, influenza vRNP may outcompete FIP2 and decrease the amount of dynein on Rab11A-containing endosomes (Fig. 8). However, further studies are needed to clarify this hypothesis. In addition, lack of direct interaction between KIF13A and Rab11A in A549 cells suggests that dynein dysregulation and not KIF13A, as previously suggested[53], may be responsible for disrupting Rab11A intracellular movement during IAV infection.

Our two-color imaging studies using PA::mRuby have demonstrated co-transport between Rab11A and IAV vRNP

segments. However, we have also observed that >50% of the IAV puncta move independently of Rab11A spots and vice versa. Thus, our current studies provide strong evidence for a microtubule and Rab11A-independent mechanism of IAV vRNP transport. We observed that transport of GFP-Rab11A foci is primarily directed on microtubules, while a significant portion of the PA::GFP motion, used as a surrogate for vRNP, is retained in the absence of intact microtubule filaments (Supplementary Movies 4 and 7). Previously we reported a decrease in the colocalization of Rab11A and influenza virus RNA in the absence of intact microtubule filaments[22], confirming the importance of microtubules in Rab11A-dependent transport of IAV vRNP. The robust motion of PA::GFP in cells treated with nocodazole provides convincing evidence for the presence of a microtubule-independent vRNP transport mechanism. Taken together with the significant decrease in GFP-Rab11A motion in nocodazole-treated cells, our data suggest that this alternative mechanism of transport may also be Rab11A-independent. Recent data using electron microscopy have implicated Rab11A-carrying irregularly coated vesicles budding from the endoplasmic reticulum (ER) as a scaffold for vRNP transport and assembly[57].

The ER is a very large cellular organelle that has recently been shown to undergo rapid oscillatory movements[58], and can sample over 90% of the cellular volume within 10 min independent of microtubule filaments[59]. Based on these previous data and our current work, we speculate that intracellular transport and assembly of IAV vRNP segments requires a membrane scaffold either in the form of Rab11A-RE or the ER, and this membrane promiscuity may provide an advantage for vRNP segment–segment assembly. Recent work has also suggested that vRNP exits the ER in liquid-phase organelles[60]. Therefore, additional work examining the intracellular movement of individual vRNP segments and ER membranes within a single cell is needed to elucidate the importance of dynamic membranous scaffolds during IAV infection.

## Methods

**Cells and viruses.** Human lung adenocarcinoma (A549 from ATCC) cells used in imaging experiments were maintained in high-glucose Dulbecco's Modified Eagle Medium (DMEM) supplemented with 10% fetal bovine serum, 2% L-glutamine, and 1% penicillin/streptomycin. A549 cells stably expressing GFP-Rab11A protein were generated by transfecting regular A549 cells with GFP-tagged Rab11A plasmids containing antibiotic resistance for Geneticin (G-418) (the plasmids were a kind gift from Dr. Ora Weisz from the University of Pittsburgh School of Medicine). Transfected cells were serially passaged at very low dilutions (average 1 cell per well in 96-well plates) in the presence of G-418 and selected based on the presence of fluorescence at every passage. The resulting stable cells (A549 GFP-Rab11A) that had incorporated GFP-Rab11A into their genome were stored at −80 °C in FBS/DMSO (90%/10%) suspension media for future use. One day prior to imaging, A549 cells were seeded directly onto 24 × 50-mm glass coverslips (#1.5 thickness) in 10-cm tissue culture dishes such that the cells formed a confluent monolayer the next day. On the day of imaging, the coverslips were washed in

phosphate-buffered saline (PBS) and mounted in stainless-steel diSPIM imaging chambers (Applied Scientific Imaging, Oregon, USA)[24]. The mounted cells were inoculated with viruses and/or treated with drugs for the requisite amount of time in the stainless-steel inserts before imaging.

Recombinant H1N1 Influenza A viruses were rescued in MDCK (obtained from ATCC) using pHW2000-based 8-plasmid system for lab-adapted strain A/WSN/33 (called WSN in this report), a kind gift from Dr. Richard Webby (St. Jude). The modified pandemic strain A/CA/07/2009 that expressed a GFP-tagged PA protein (the virus is called H1N1pdm PA::GFP in this report) was rescued in MDCK cells (ATCC). Fluorescent viruses were further selected by picking individual plaques, growing them further in specific pathogen-free chicken eggs (Charles River Laboratories, MA, USA), infecting MDCK cells, and selecting clones that expressed GFP in more than 90% of infected cells[7,61]. Recombinant H1N1 virus A/WSN/33 expressing mRuby-tagged PA protein (the virus is called PA::mRuby in this report) was rescued using the same strategy used for PA::GFP[7,61]. The GFP-tagged viral protein PA (called PA::GFP in this report) is a good surrogate for studying the motion of IAV vRNP, since over 90% of PA::GFP puncta within A549 cells infected with H1N1pdm PA::GFP colocalize with the viral NP (Supplementary Fig. 5a). H1N1pdm PA::GFP virus showed growth in A549 cells in the presence of nocodazole and latrunculin A that was indistinguishable from DMSO (see Supplementary Fig. 5b). Therefore, using these drugs provides a reasonable assay to examine the transport dynamics of vRNP in the absence of microtubule or actin filaments. To allow for robust expression of GFP signal, all imaging of H1N1pdm PA::GFP infections were performed at 16 hpi. RFP-expressing RSV was a generous gift from Dr. Mark Peeples at The Ohio State University, College of Medicine and Dr. Peter Collins at the NIH[62]. IBV, B/Texas/02/2013 (Victoria Lineage), Cell-Derived, FR-1302, was obtained through International Reagent Resource, Influenza Division, WHO Collaborating Center for Surveillance, Epidemiology and Control of Influenza, and Centers for Disease Control and Prevention, Atlanta, GA, USA.

**Drug treatments**. A549 cells were treated with the microtubule-depolymerizing drug nocodazole at a concentration of 16 μM (nocodazole mol. wt. = 301.3 g/mole, 5 mg/mL solution in dimethyl sulfoxide (DMSO) diluted to 1:1000). Similarly, the concentration of actin- depolymerizing drug, latrunculin A, was fixed at 1 μM. To study the effect of drugs on viral transport, the cells were inoculated with virus and the infection allowed to proceed for 4 h before commencing drug treatment to allow unhindered virus attachment, internalization, transport, and replication in the nucleus. The cells were exposed to the drugs for a minimum of 4 h prior to imaging. The drug doses were determined based on published literature as discussed in our previous report.

**Viral infections and growth assays**. Influenza A and B virus infections were performed as noted previously[22]. In brief, influenza stocks were diluted in complete MEM (Minimum Eagle Medium (MEM) + 2% L-glutamine + 2% antibiotic/anti-mycotic) supplemented with 1:1000 TPCK-trypsin (for all influenza viruses except for WSN virus). A total volume of 0.5-mL inoculum was prepared at the requisite MOI for $2 \times 10^6$ cells. The cells were washed in PBS, mounted in diSPIM imaging chamber inserts, and incubated at room temperature for 1 h to aid virus attachment. The inoculum was removed and replaced with complete MEM-TPCK-trypsin and the imaging chamber was placed in the incubator. RFP–RSV stocks (RSV-A2 strain expressing free reporter red fluorescent protein (RFP), originally from Dr. Mark Peeples [Ohio State University] and Dr. Peter Collins [NIH]) were diluted in OptiMEM (with 2% FBS, 1:1000 L-glutamine, and 1:1000 antimycotic–antibiotic) to an MOI = 0.1 pfu/cell. Cells were incubated as described above with inoculum at 37 °C with intermittent rocking every 15 min for 2 h. The inoculum was then replaced with complete MEM and cells placed back in the incubator and imaged at 16 hpi.

The effect of nocodazole and latrunculin A on H1N1pdm PA::GFP virus replication was determined by measuring viral titers by tissue culture infectious dose 50 (TCID$_{50}$) using the endpoint titration method[63]. Briefly, A549 cells were infected with H1N1pdm PA::GFP virus, treated with drugs at 4 hpi, and the cell supernatant was collected at 8, 16, and 24 hpi for performing viral titration. Viral titers were reported as log$_{10}$ TCID$_{50}$/mL (Supplementary Fig. 5b).

**Immunofluorescence, FISH, and image analysis in fixed cells**. Protocols used for imaging and colocalization analysis in fixed, immunofluorescence-, or FISH-stained cells have been detailed in previous publications[7,22]. Briefly, for immuno-fluorescence, cells were fixed using 4% paraformaldehyde (PFA) for 10 min, followed by thorough washing with PBS. Fixed cells were incubated with 0.5% Triton X-100 solution for 10 min for permeabilization. Fixed, permeabilized cells were blocked with 3% bovine serum albumin (BSA, fraction V) in PBS for 1 h, incubated with the primary antibody in 1% BSA in PBS for 1 h, washed, and finally incubated with the secondary antibody in 1% BSA in PBS for 1 h. Influenza H1N1 NP proteins were immunostained and imaged using mouse antibody HB-65 (IRR, 1:1000 dilution), while influenza B NP protein was identified using Abcam antibody #20711 (1:500 dilution). Antibodies for staining the following cellular components were sourced as follows: microtubules (mouse monoclonal tubulin antibody T-5168—Sigma, 1:2000 dilution), actin (phalloidin conjugated to Alexa-Fluor 594 Invitrogen/Thermo Fisher #A12381, 1:40 dilution). Nuclear DNA was

stained using DAPI (4′,6-diamidino-2-phenylindole) obtained from Thermo Fisher (#D1306, 1:5000 dilution). WSN PA vRNA was imaged using FISH using probes from Biosearch Technologies. Briefly, infected cells were fixed at 8 hpi with 4% PFA and stored in ice-cold 70% EtOH overnight. The next day the coverslips were rehydrated with the wash buffer (10% formamide and 2× SSC in DEPC-treated water). Cells were incubated overnight in 200 ml of hybridization buffer (10% dextran sulfate, 2 mM vanadyl-ribonucleoside complex, 0.02% RNA-free BSA, 1 mg/ml E. coli tRNA, 2× SSC, and 10% formamide in DEPC-treated water) with 2 ml of labeled probes (probe concentrations ranged from 2.5 to 10 mM) and in a 28 °C incubator. The following day, the hybridization buffer was removed and cells were incubated in wash buffer containing DAPI for 10 min, washed, and then mounted. All incubations were performed in the dark. Immunostained and FISH samples were mounted using Prolong Diamond anti-fade mountant (Thermo Fisher # P36970) and allowed to cure overnight. Imaging was performed on Olympus Fluoview 1000 confocal microscope with a 60×, 1.35-NA oil objective (X–Y pixel size = 51 nm, Z spacing = 170 nm). The resulting images were deconvolved using Huygens software (SVI, Netherlands). Locations of proteins or viral RNA of interest were identified using "Spots" function in Imaris (Bitplane Inc.) and subsequent colocalization analysis was performed using custom MATLAB-based scripts (XTensions) in Imaris freely available for download at https://github.com/Lakdawala-Lab/MatLab-Extensions. The script considers a spot from channel A to be colocalized with channel B if they are found within 300 nm (user selectable) of each other and alone otherwise. The degree of colocalization is calculated as follows:

$$\% \, \text{Colocated} = \frac{N_{\text{coloc}}}{N_A + N_B}, \tag{1}$$

where $N_{\text{coloc}}$ is number of spots of A (or B) found colocalized, $N_A$ is the total number of spots in channel A, and $N_B$ is the total number of spots in channel B. Statistical comparisons were performed in Graphpad Prism using two-way ANOVA with multiple comparisons. All tests used in this report are two-sided.

**Light-sheet microscopy and image analysis**. The protocols used for diSPIM imaging and subsequent analysis used by our lab have been recently published[64]. Briefly, A549 cells were seeded on glass coverslips and subsequently treated with cytoskeletal drugs and/or infected as described above. Thirty minutes prior to imaging, the cells are washed with PBS and the cell media was replaced with low-background-fluorescence imaging media (Fluorobrite DMEM, Gibco) to flush out any phenol red dye used in cell-culture media, which can generate background fluorescence. Live-cell microscopy was performed on a homebuilt dual-view inverted selective-plane illumination microscope (diSPIM) equipped with a heated stage maintained at 37 °C[23]. The microscope was built and aligned using protocols detailed previously[24]. Briefly, the bulk of the optomechanical hardware, automated stages, laser scanners, piezo elements for focus control, and control electronics were purchased from Applied Scientific Instruments (Eugene, Oregon, USA). Each arm of the diSPIM consists of a water-dipping objective (Nikon, MRD07420, 40×, 0.8 NA) that relays the image to a scientific complementary metal–oxide semiconductor camera (Hamamatsu Flash4v2 Orca) capable of imaging our field of view at 200 Hz. Two semiconductor solid-state lasers (488 nm, 100 mW and 561 nm, 150-mW Excelsior laser, Spectra Physics, CA, USA) are combined with a dichroic filter (Di02-R488-25 × 36, Semrock). An acousto-optical tunable filter (AOTFnc450.600, AAopto, Orsay, France) selects the wavelength and average power of excitation. The lasers are fiber-coupled to the diSPIM head laser scanners. The lasers are reflected to the objectives and the return fluorescence is transmitted by a notched dichroic (ZT405/488/561/IR RPC, Chroma). High-speed digital-to-analog converter cards (PXI-6733, National Instruments) control the timing and waveforms for moving the piezos, scanners, AOTF, and triggering camera capture. Custom code written in LabView is used for hardware interfacing, data capture, and storage.

The diSPIM resolution was characterized using 100-nm fluorescent beads (Thermo Fisher F8803) embedded in 0.2% agarose gel. Full width at half-maximum (FWHM) numbers were calculated for ~30 beads along all (x-, y-, and z-) axes before and after joint deconvolution (discussed below). Supplementary Fig. 1 shows sample bead profiles and FWHM distribution. Before deconvolution, bead FWHM is denoted here along the following axis (x: 0.4 ± 0.03 μm, y: 0.4 ± 0.03 μm, and z: 1.4 ± 0.2 μm). Post deconvolution, the bead FWHM for the same samples is found to be approximately isotropic (x: 0.27 ± 0.03 μm, y: 0.31 ± 0.03 μm, and z: 0.36 ± 0.03 μm). FWHMs are listed in Supplementary Table 1 and were measured using the ortho-slicer tool in Huygens deconvolution software (Scientific Volume Imaging B.V., Netherlands). Light-sheet thicknesses were measured by focusing the objectives on 100-nm fluorescent beads, while scanning the light sheets through the beads. The light-sheet thickness FWHM was found to be ~1.5 μm at the center, ~4 μm at 20 μm to each side of the focus.

Volumetric images were obtained at ~1.4 Hz with 50 slices per volume (0.5-μm spacing) and 512 × 512 pixels per slice (162.5-nm pixel width and height) using the following range of powers at the sample focus—488-nm power: 40–60 μW, 561-nm power: 70–120 μW. Two-color imaging was performed by sequentially imaging GFP and mRuby signals. The addition of blank frames between acquisition of the two colors was deemed necessary to reduce cross-excitation between channels. This modified imaging process resulted in datasets with time resolution (frame-to-frame time interval) of ~2.8 s compared with

single-color datasets with a time interval ~0.7 s. Dual-view volumetric images were cropped and background subtracted in ImageJ (NIH). The two views (referred to as SPIM$_A$ and SPIM$_B$) are co-registered and deconvolved jointly to obtain a single volumetric image stack with an isotropic voxel spacing of 162.5 nm and isotropic spatial resolution. Image registration and deconvolution are performed on a graphics-processing unit (GPU, NVIDIA K6000) using custom software written in C++/CUDA using Visual Studio 2013 and CUDA toolkit v7.5 for GPU cards with compute capability 1.0 or higher[65] (code published by Min Guo and Hari Shroff, NIH at https://doi.org/10.1101/647370). The software first transforms SPIM$_B$ (i.e., rotated, translated, scaled, and skewed using 12 total degrees of freedom) to overlay with SPIM$_A$. The quality of registration is optimized by minimizing a cost function defined using a cross-correlation of the two images[66]. Minimization of the cost function is performed using Powell's search method for a minimum (http://mathfaculty.fullerton.edu/mathews/n2003/PowellMethodMod.html). After calculating the best transformation matrix for registration, the images are deconvolved jointly using a modified Richardson–Lucy deconvolution algorithm described previously[23].

Visualization was performed using Imaris 8.4.2 (Bitplane, Inc.), which was used to identify single cells and isolate their fluorescence signal isolated using the manual surface generation function (see Supplementary Movie 1). During analysis of RFP–RSV-infected cells, only the trajectories of Rab11A within RFP-positive cells were isolated using manual surface generation, then quantified, and analyzed for statistics (see Supplementary Fig. 9). In addition, for tracking fluorescent influenza vRNP, the nuclear volume is delineated using a second manually generated surface and the signal inside ignored. Fluorescent puncta were identified using Imaris built-in spot generation algorithm (with a seeding spot size of 330 nm). Step-by-step protocols detailing the procedure have been previously published[64]. Briefly, spot generation was achieved manually for each dataset using intensity at the center of each spot as the threshold. The thresholds were lowered till each moving signal density was assigned a spot. The spots were tracked over time to generate motion statistics for each cell. Based on the resolution measurement, we cannot distinguish individual fluorescent emitters within a diffraction-limited sphere of diameter ~300 nm. Thus, each identified spot in our dataset may contain multiple GFP molecules, and hence multiple Rab11A proteins and multiple IAV vRNP segments. By filtering out tracks smaller than three steps, we can eliminate counting of spurious spots that arise due to thermal or dark-current noise in the camera. Supplementary Fig. 13 shows that the spots selected for tracking do not show a different distribution of intensity (at the spot center) from that for all detected spots. Thus, no inherent bias exists toward selecting brighter spots over dimmer ones. Track length, displacement, duration, speed, and normalized standard deviation of speed are calculated per track using built-in routines in Imaris. These parameters are used to further calculate track straightness, arrest coefficients, and velocities using MATLAB plugins written for Imaris or by exporting the data out of Imaris and then using dedicated MATLAB scripts (analysis scripts freely available at https://github.com/Lakdawala-Lab/MatLab-Extensions/). The exact definitions of various parameters used for analyzing intracellular transport are detailed in the "Results" section. To ensure that changes observed during drug treatments or viral infections were specific to altered movement dynamics, we also exported and compared the total number of identified spots and tracks from each cell between experimental conditions. We found the number of spots (200,000–600,000) and tracks (20,000–50,000) to be fairly consistent across treatments.

To study the effect of drug treatments and/or viral infections on cellular transport in a statistical manner, while still recognizing the cell-to-cell variation within each treatment, we devised the following protocol: the various cellular transport parameters exported from Imaris were binned into a normalized histogram, one per measure per cell (illustrated in Fig. 1). To further distill the information corresponding to a given transport parameter (e.g., track displacement) from tens of thousands of tracks to a single number for a cell, we plotted the cumulative histogram. In the case of track displacement, the value of displacement at which the cumulative frequency achieves 50% probability is defined as the CF$_{50}$ of displacement for that cell and is determined manually. The CF$_{50}$ determinations are performed using custom MATLAB scripts, which are freely available at our lab's GitHub site (https://github.com/Lakdawala-Lab/MatLab-Extensions/).

CF$_{50}$ values are used to describe the behavior of cellular transport for the whole cell. To compare the effect of treatments, statistical comparisons between CF$_{50}$ were performed in Prism software (Graphpad Inc.) using a one-way ANOVA algorithm by comparing the mean of each treatment with the means of other treatments for each parameter (also see "Analyzing three-dimensional movement at single-cell level" of Results). All tests used in this report are two-sided.

**Immunoprecipitation assay**. A549 GFP-Rab11A cells were infected at an MOI of 1 for 16 h, at which point 70% of the cells were infected (as determined by immunofluorescence). Cells were solubilized with NP40 lysis buffer (50 mM Tris, pH 7.4, 150 mM NaCl, 0.5 mM EDTA, and 0.5% NP40) and 1 mg of protein was immunoprecipitated with anti-GFP magnetic beads overnight as described by the manufacturer (MLB). This assay and subsequent mass spectrometry and western blot analysis were repeated twice to obtain two independent biological replicates.

**Western blot analysis**. Proteins were separated by sodium dodecyl sulfate polyacrylamide gel electrophoresis (SDS-PAGE) and transferred to a nitrocellulose membrane (Bio-Rad). Membranes were probed with mouse anti-dynein (Invitrogen), rabbit anti-GFP (Living Color), or mouse anti-PB2 (BEI) primary antibodies at a dilution of 1:1000 and the appropriate horseradish peroxidase-conjugated secondary antibodies (Jackson Laboratories) used at a dilution of 1:4000. Proteins were detected using Pierce ECL Western Blotting substrate (Thermo Scientific). For quantitation, the pixel intensity for each band was determined using the ImageJ program (NIH) and then normalized to the indicated control. The original uncropped and unprocessed images of the western blot gels have been included in the Source Data file.

**Sample preparation for proteomics analysis**. All chemicals used for preparation of nanoflow liquid chromatography–tandem mass spectrometry (nLC–MS/MS) samples were of sequencing grade and purchased from Sigma-Aldrich (St. Louis, MO), unless otherwise stated. The Rab11A IP eluate was separated by SDS–PAGE using NuPAGE 1DE System (NuPAGE Novex 4–12% bis–tris 1.5-mm gels, Thermo Fisher Scientific, Waltham, MA). Visualization of separated proteins was performed by overnight staining with Coomassie blue G-250 solution (Thermo Fisher Scientific, Waltham, MA). The in-gel tryptic digestion followed by peptide extraction from the gel bands was performed according to previously published protocols[67]. The extracted peptides were desalted using Poros Oligo R3 RP (Perseptive Biosystems, Framingham, MA) P200 columns with C18 3 M plug (3 M Bioanalytical Technologies, St. Paul, MN) prior to nLC–MS/MS analysis. The Rab11A IP input samples were processed using the suspension trap (S-Trap, Protifi, Huntington, NY)[68] mini spin column digestion protocol with minor modifications. The Rab11A IP input samples were first mixed with 20% SDS to the final concentration of 5% SDS following the manufacturer's procedure. The peptide solution was pooled, lyophilized, and desalted prior to nLC–MS/MS.

**Nanoflow liquid chromatography–tandem mass spectrometry**. The peptide mixture was separated using a Dionex Ultimate 3000 high-performance liquid chromatography (HPLC) system (Thermo Scientific) equipped with a two-column setup, consisting of a reversed-phase trap column (Acclaim PepMap100 C18, 5 µm, 100 Å, 300 µm i.d. × 5 mm, Thermo Scientific) and a reversed-phase analytical column (30 cm, 75 µm i.d., 360 µm o.d., packed with Pur C18AQ, 3 µm; Dr. Maisch). Loading buffer was 0.1% trifluoroacetic acid (Merck Millipore) in water. Buffer A was 0.1% formic acid, and Buffer B was 80% acetonitrile + 0.1% formic acid. The HPLC was coupled online with an Orbitrap Fusion mass spectrometer (Thermo Scientific, San Jose, CA). The gradient was 65 min from 4 to 35% buffer B at a flow rate of 300 nl/min for Rab11A IP samples, and 120 min for Rab11A IP input samples. The MS instrument was controlled by Xcalibur software (Thermo Fisher Scientific). The nanoelectrospray ion source (Thermo Scientific) was used with a spray voltage of 2.2 kV. The ion transfer tube temperature was 275 °C. Data acquisition was performed in the Orbitrap for both precursor and product ions. MS survey scans were obtained for the m/z range of 350–1200 in the Orbitrap with maximum ion injection time of 50 ms, auto gain control target $5 \times 10^5$, and a mass resolution of 120,000. MS/MS was performed with a TopSpeed duty cycle set to 3 s. Higher collisional dissociation was set to 30. MS/MS was acquired in the Orbitrap for the Rab11A IP samples, and in the ion trap for the Rab11A IP input samples.

**Protein identification and quantification**. The raw mass spectrometer files were processed for protein identification using the Proteome Discoverer (v2.3, Thermo Scientific) and the Sequest HT algorithm with a peptide mass tolerance of 10 ppm, fragment m/z tolerance of 0.025 Da for Rab11A IP MS files and 0.25 Da for Rab11A IP input files, and a false-discovery rate (FDR) of 1% for proteins and peptides. All peak lists were searched against the UniProtKB/Swiss-Prot database of Human (March 2019, 20417 entries) and UniprotKB/TrEMBL Influenza A virus (A/California/07/2009(H1N1); March 2019, 10 entries) sequences using the parameters as follows: enzyme, trypsin; maximum missed cleavages, 2; fixed modification, carbamidomethylation (C); variable modifications, oxidation (M), protein N-terminus acetylation. Peptide quantifications were log$_2$ transformed and normalized using the median of the distribution for each sample. Missing values were imputed using a distribution of values of 30% width and two standard deviations lower than the average of the distribution of valid values. Statistical difference between the two conditions was assessed using a peptide-based paired $t$ test $p$ value; briefly, we first calculated the fold-change regulation of all peptides belonging to a given protein, and then the $p$ value was assessed using the reproducibility of those fold changes (significant if statistically different from no change; $t$ test $p$ value < 0.05). Protein fold changes were obtained by averaging the fold-change regulations of all peptides belonging to a given protein. GO biological process information ranked by $p$ value enrichment score was obtained from GeneGo's MetaCore pathways analysis package (Thomson Reuters) with FDR < 0.01.

**Reporting summary**. Further information on research design is available in the Nature Research Reporting Summary linked to this article.

## Data availability

Data underlying Figs. 3, 4, 5, 6, 7d, Supplementary Figs. 2, 4, 5, 8, 10–12 are provided as Source Data files. Data underlying Supplementary Fig. 1 are included as Supplementary Table 1. Data underlying Fig. 7b, c are provided as Supplementary Data 1 file. Mass spectrometry proteomics data have been deposited to ProteomeXchange Consortium (http://proteomecentral.proteomexchange.org) via the PRIDE partner repository[69] with the dataset identifier PXD016331 or at https://www.ebi.ac.uk/pride/archive/projects/PXD016331. All of the raw diSPIM imaging data are available as open-source TIFF files on FigShare at https://doi.org/10.35092/yhjc.c.4719353. All statistical results are provided as Supplementary Data 2 file. All other data are available from the corresponding author upon reasonable requests.

## Code availability

The computer code used to acquire images is written in LabView, the code for cropping and image processing is written using open-source imageJ macro language, and CF₅₀ analysis and intensity renormalization are performed using custom MATLAB code. All of the above are available on our lab GitHub at https://github.com/Lakdawala-Lab. The dual-view registration and deconvolution are performed using custom code written in C++/CUDA, which is published at https://www.biorxiv.org/content/10.1101/647370v1 and is available from Hari Shroff on request. Spot recognition and tracking are performed in commercially available Imaris software. Colocalization and arrest coefficient analysis are performed using custom MATLAB extensions to Imaris also available at https://github.com/Lakdawala-Lab. Statistical tests and graphs were made using commercially available Graphpad Prism version 8 software. Graphing and statistics for violin plots were made using Python's open-source Seaborn package [https://seaborn.pydata.org/].

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

## Acknowledgements

We thank Dr. Yicong Wu and Dr. Abhishek Kumar for many helpful discussions and suggestions, Dr. Ora Weisz for providing us with GFP-Rab11A and mCherry-Rab11A constructs and helpful discussions, Dr. Jennifer Bomberger and her lab for technical assistance with RSV, Haley Cartwright for technical troubleshooting, and Kyle Grabowski and Logan Hellinger for assistance with image analysis and movie editing. This work was funded by NIH grant (1K22AI108600-01A1 and 1R01AI139063-01A1), the American Lung Association Biomedical Research grant, and a New Initiative Award from the Charles E. Kaufman Foundation, a supporting organization of The Pittsburgh Foundation. This work was also supported by the National Institutes of Health (NIH) through the NIH Director's New Innovator Award Program (1-DP2-A1112243), and the intramural research program of the National Institute of Biomedical Imaging and Bioengineering (NIBIB).

## Author contributions

A.R.B. and S.S.L. conceptualized and devised the studies. A.R.B., V.L., E.N., and J.J. performed cell-culture-based experiments and rescued viruses. A.R.B. built the diSPIM and developed the image analysis pipeline. K.K., B.G., and M.W. performed mass spectrometry and subsequent analysis. M.G. and H.S. provided technical guidance and accelerated GPU/CUDA code. E.T.K. performed fractionation western blots. A.R.B. and S.S.L. wrote the paper with input from all authors.

## Competing interests

The authors declare no competing interests.
