## [Peer Review File · Nature Communications]

Reviewers' comments:

Reviewer #1 (Remarks to the Author):

Influenza A viruses (IAVs) have a segmented RNA genome which is replicated in the nucleus of infected cells. It is poorly understood how the newly synthesized viral RNA segments (vRNAs), in the form of viral ribonucleoproteins (vRNPs), are transported from the nucleus to the plasma membrane and how they assemble into bundles of 8 different vRNA segments to be incorporated into budding progeny virions. There is evidence for a role of the Rab11 GTPase in these processes, but how Rab11 is mechanistically involved remains unclear. This study by Bhagwat et al. expands upon previous work from the same team, which aimed to investigate the role of microtubules in influenza viral RNA (vRNA) transport and assembly (Nturibi et al, Journal of Virology 2017). Their initial study relied on immunofluorescence colocalization analysis and fluorescence in situ hybridization on fixed cells. Here the authors used high spatiotemporal resolution light-sheet microscopy to characterize Rab11A motion in live cells during infection with an IAV or a Respiratory Syncytial virus (RSV), as well as influenza vRNP movements, in the presence or absence of drugs that specifically disrupt the microtubule or actin cytoskeleton filaments. Based on the obtained images, the authors conclude that i) Rab11A motion is primarily dependent on microtubules and not actin in uninfected cells, ii) infection with IAVs slows down Rab11A motion and promotes more stalled motion, whereas infection with RSV alters Rab11A motion in the opposite way iii) influenza vRNPs motion is essentially independent from microtubules.

Overall this is a very well-presented study, in which the imaging data support the conclusions drawn by the authors. The workflow is well explained and convincing. The quality of the live imaging data is impressive. This study contributes to advance the understanding of influenza virus vRNA intracellular transport. However, the advance would be much more significant if additional experiments were performed to provide more mechanistic insights.

(1) The presented data raise the possibility that vRNP transport is independent of the bright GFP-Rab11A foci, which does not necessarily mean “independent of Rab11A”, as discussed by the authors. However evidence is indirect: it relies mostly on the observation that upon nocodazole treatment, the motion of GFP-Rab11A foci is decreased, the colocalization of PAVRNA and Rab11A decreases, whereas the motion of PA::GFP puncta remains unaltered. However, as shown in the

Supplementary Figure 3C, about 40% of total PAVRNA and Rab11A spots show colocalization in untreated cells.

Evidence would be much stronger if the tracking of Rab11A and PA was performed simultaneously in live cells. Here the authors used alternatively a GFP-Rab11 cell line and a PA::GFP virus. Is there any technical limitation to the use of a two-colour experimental setting ?

(2) Previously published work suggest that KIF13A, a member of the kinesin-3 family, mediates influenza vRNP transport (Ramos-Nascimento et al, J Cell Sci 2017) and that vRNPs compete with Rab11 FIP effectors (Vale-Costa et al, J Cell Sci 2016). Investigating the effect of KIF13A depletion/overexpression and FIP overexpression on Rab11A and vRNP motion would provide further mechanistic information.

Other comments.

(3) Figure 1 : the workflow is well explained, as well as how the CF50 values will be compared to evaluate the effect of different experimental conditions. Consistently, the legend of Supplementary Figures 2, 6 and 8 refer to the CF50 : « Each data point represents an individual analyzed cell using CF50 ». For some reason, it is not the case in the legends of the main Figures 3, 4, 5 and 6. They refer to « mean speed » « mean velocity », which is confusing.

(4) In Figure 2 and Supplementary Figure 5, the size of « GFP-Rab11 foci » and « PA::GFP puncta » is in the same range (0.5 – 1 μm in diameter). In the discussion the authors raise the possibility that the bright GFP-Rab11 foci may represent Rab11 associated with recycling endosomes. Would this imply that the PA ::GFP puncta may represent groups of vRNPs ? and/or vRNPs associated with transport vesicles ? this point should be discussed.

(5) Supplementary Figure 3B and page 19. The statement « H1N1pdm PA ::GFP virus showed robust growth in A549 cells... » is inaccurate, as the PA ::GFP virus grows to very low titers compared to its wild-type virus counterpart.

(6) Supplementary Figure 3C. The numbers of counted « PA vRNA » and « Rab11A » spots should be indicated.

(7) When discussing the opposite effect of IAV and RSV infection on Rab11A motion, the authors should not exclusively refer to the segmented versus non-segmented nature of their genome, and be more comprehensive.

(8) page 7. « These data are consistent with increased microtubule-based movement in the absence of the actin cytoskeletal network ». It is unclear whether this a statement based on available literature (in this case the appropriate references should be added) or a speculation.

(9) WSN/1933 or A/WSN/1933 should be spelled A/WSN/33.

(10) The references 1 and 2 are inappropriately numbered.

Reviewer #2 (Remarks to the Author):

OVERVIEW:

Several viruses have been shown to use and manipulate Rab11 during their lifecycles (as reviewed in (Bruce et al. 2012)). In this manuscript, the authors have used light sheet microscopy approaches to generate 3D images of the cell and characterize what happens to Rab11 in the presence of cytoskeleton acting drugs and infection with influenza A and B viruses (IAV, IBV, respectively) as well as with respiratory syncytial virus (RSV). The manuscript confirms that RSV increases the movement of Rab11 whilst IAV leads to stalling of Rab11 movement (Amorim et al. 2011; Avilov et al. 2012a; Avilov et al. 2012b; Chambers & Takimoto 2010; Einfeld et al. 2011; Kawaguchi et al. 2015a; Momose et al. 2011; Utley et al. 2008; Vale-Costa et al. 2016). Similarly to others, the authors hypothesize that the differential modulation of Rab11 during infection relates to the nature of their viral genomes being non-segmented (RSV) or segmented (IAV, IBV), but do not prove this assumption. Conversely to having to transport one segment for incorporation in a virion as RSV, IAV and IBV require that the eight different segments, that constitute the viral genome, form a supra-molecular complex to assemble fully infectious particles. The manuscript presents an interesting methodology to quantify and statistically analyze vesicular trafficking but it does not represent a significant advancement in the knowledge of how Rab11 is manipulated during the different viral infections.

MAJOR ISSUES:

- The study here presented does not provide compelling mechanistic and conceptual advance. A number of previous studies have indicated that IAV viral ribonucleoproteins (vRNPs) associate with Rab11A after their delivery to the cytoplasm in late-stage infection (Amorim et al. 2011; Avilov et al. 2012a; Avilov et al. 2012b; Momose et al. 2011). It was also repeatedly shown that infection altered Rab11A in a number of ways including cellular distribution (from small to big foci), binding to FIPs and to microtubules (Amorim et al. 2011; Avilov et al. 2012a; Avilov et al. 2012b; Eisfeld et al. 2011; Kawaguchi et al. 2015a; Momose et al. 2011; Vale-Costa et al. 2016). Recent manuscripts have, however reinforced that RNPs use a Rab11-independent transport, binding to the endoplasmic reticulum (de Castro Martin et al. 2017) or as this author has shown, having uncoupled subcellular locations in the presence of nocodazole (Nturibi et al. 2017). The stalling of Rab11 movement during influenza A infection has also been shown, namely competition between FIPs and RNPs for Rab11 binding (Kawaguchi et al. 2015b; Vale-Costa et al. 2016). All the data has led to the proposal that the virus creates a platform to concentrate the eight different RNPs to promote the formation of supramolecular complexes.

In relation to paramyxoviruses, it was reported that Sendai virus and Respiratory syncytial virus use Rab11. In the case of RSV however, it was demonstrated that the virus relied on FIP2 and binding of the genome did not compete with FIP adaptor. Therefore, the Bhagwat paper confirms previously reported data, but does not add mechanistic insight.

- All the graphs show that uninfected and untreated cells have a wide dispersion of the mean values calculated. 10 cells were analyzed for this condition. However, for all other treatments the number of the cells analyzed was lower, varying from 4-6 cells. The case of Latrunculin A treatment for example of Fig 2, the cells have a bimodal behavior, with some being affected by the treatment and others not. Analyzing the same number of cells is preferential, especially considering the dispersion of untreated samples.

- The number of moving Rab11/RNPs is very different between conditions. Bigger aggregates are brighter than the smaller versions, which will bias the analyses and acquisition of moving targets. This is particular clear when analyzing S6-S8. Please comment. The inclusion of the number of moving vesicles would complement the data. Also, actin/microtubule-based movements have very different behaviors, and different cellular localization. The methodology described would allow discriminating movements relatively to their cellular location - close to the nucleus or to the plasma membrane, which would be an insightful way to look at Rab11 positioning during viral infection, showing novel and relevant alterations to this molecule for IA, IB and RSV.

- Videos for direct comparison are very different. This includes the set Supplementary movie 6 to Supplementary movie 8, in which Sup movie8 is formatted in a very distinct manner and does not allow direct comparison.

- Representative Movies for Figure 6 have not been included.

REFERENCES

Amorim, M. J., Bruce, E. A., Read, E. K., Foeglein, A., Mahen, R., Stuart, A. D., & Digard, P. 2011. A Rab11- and microtubule-dependent mechanism for cytoplasmic transport of influenza A virus viral RNA. *J Virol*, 85(9): 4143-4156.

Avilov, S. V., Moisy, D., Munier, S., Schraidt, O., Naffakh, N., & Cusack, S. 2012a. Replication-competent influenza A virus that encodes a split-green fluorescent protein-tagged PB2 polymerase subunit allows live-cell imaging of the virus life cycle. *J Virol*, 86(3): 1433-1448.

Avilov, S. V., Moisy, D., Naffakh, N., & Cusack, S. 2012b. Influenza A virus progeny vRNP trafficking in live infected cells studied with the virus-encoded fluorescently tagged PB2 protein. *Vaccine*, 30(51): 7411-7417.

Bruce, E. A., Stuart, A., McCaffrey, M. W., & Digard, P. 2012. Role of the Rab11 pathway in negative-strand virus assembly. *Biochem Soc Trans*, 40(6): 1409-1415.

Chambers, R., & Takimoto, T. 2010. Trafficking of Sendai virus nucleocapsids is mediated by intracellular vesicles. *PLoS One*, 5(6): e10994.

de Castro Martin, I. F., Fournier, G., Sachse, M., Pizarro-Cerda, J., Risco, C., & Naffakh, N. 2017. Influenza virus genome reaches the plasma membrane via a modified endoplasmic reticulum and Rab11-dependent vesicles. *Nat Commun*, 8(1): 1396.

Eisfeld, A. J., Kawakami, E., Watanabe, T., Neumann, G., & Kawaoka, Y. 2011. RAB11A is essential for transport of the influenza virus genome to the plasma membrane. *J Virol*, 85(13): 6117-6126.

Kawaguchi, A., Asaka, M. N., Matsumoto, K., & Nagata, K. 2015a. Centrosome maturation requires YB-1 to regulate dynamic instability of microtubules for nucleus reassembly. *Sci Rep*, 5: 8768.

Kawaguchi, A., Hirohama, M., Harada, Y., Osari, S., & Nagata, K. 2015b. Influenza Virus Induces Cholesterol-Enriched Endocytic Recycling Compartments for Budozone Formation via Cell Cycle-Independent Centrosome Maturation. *PLoS Pathog*, 11(11): e1005284.

Momose, F., Sekimoto, T., Ohkura, T., Jo, S., Kawaguchi, A., Nagata, K., & Morikawa, Y. 2011. Apical transport of influenza A virus ribonucleoprotein requires Rab11-positive recycling endosome. *PLoS One*, 6(6): e21123.

Nturibi, E., Bhagwat, A. R., Coburn, S., Myerburg, M. M., & Lakdawala, S. S. 2017. Intracellular Colocalization of Influenza Viral RNA and Rab11A Is Dependent upon Microtubule Filaments. *J Virol*, 91(19).

Utle, T. J., Ducharme, N. A., Varthakavi, V., Shepherd, B. E., Santangelo, P. J., Lindquist, M. E., Goldenring, J. R., & Crowe, J. E., Jr. 2008. Respiratory syncytial virus uses a Vps4-independent budding mechanism controlled by Rab11-FIP2. *Proc Natl Acad Sci U S A*, 105(29): 10209-10214.

Vale-Costa, S., Alenquer, M., Sousa, A. L., Kellen, B., Ramalho, J., Tranfield, E. M., & Amorim, M. J. 2016. Influenza A virus ribonucleoproteins modulate host recycling by competing with Rab11 effectors. *J Cell Sci*, 129(8): 1697-1710.

Reviewer #3 (Remarks to the Author):

This manuscript presents a study of movement of Rab11A and vRNA. In particular, the authors show the role of actin and microtubules on the intracellular transport of these proteins. They demonstrate that Rab11A movement is altered by different kinds of RNA viral infections. They propose that Rab11A-based transport is a common pathway that is regulated during virus infections. The authors use a light-sheet microscope (diSPIM) to capture fluorescently-tagged Rab11A and vRNA in live.

I will strictly limit the scope of this review to the technical aspects of the imaging results, because I am not familiar with the current field of virology.

General comments about fluorescence imaging:

The use of light-sheet microscopy is a valid approach in this study, considering quick vesicle movement within the 3D space and the need for tracking thousands of fluorescent foci over relatively long periods of time. However, I have some concerns about the robustness and preciseness of their results due to missing control experiments and a lack of quantitative data in some aspects.

1. The authors provide very little detail about the equipment and performance information for their diSPIM system. For example, which camera, lasers and objective lenses were used in the measurements? What's the actual spatial resolution? Does the system really provide an isotropic spatial resolution? How is the PSF across the FOV in different z positions? What's the light sheet thickness? What's the beam's power density at the sample plane? Because the imaging experiments were performed on a 'home-built' microscope in Dr. Lakdawala's lab, they should verify the system performance and describe the information elsewhere besides the reference [52].

2. To characterize the movement of fluorescent foci would be affected by sample preparation. GFP-tagged proteins were over-expressed and imaged in live. How does the over-expression of the proteins affect the cellular physiology? How does the presence of the GFP-tag affect the movement? The questions could have been answered by appropriate control experiments in which proteins with known movement were imaged and quantified in the same condition.

3. Vesicle tracking not only requires high-speed imaging but also a rigorous tracking algorithm. The authors used Imaris to identify locations of fluorescent foci. When looking at Figure 2 and S5, I wonder if the "Spots" function is rigorous enough. It's because some of GFP foci were unidentified or not donated by red spots. A couple of questions also arise, "Can the authors reliable identify GFP foci when they are packed?" "How is the identification affected by foci density?"

4. An artifact is observed in Movie S3, S4, S5 and S8. The shapes of GFP foci in the XZ plane are diagonally elongated (see the attached pictures). It's probably because the raw data are not so good or image registration is not optimal. With such a condition, it is difficult to evaluate the accuracy of their data.

Specific comments:

P.2: The authors stated "... (diSPIM) was developed as a tool for imaging live cells at high spatiotemporal resolution...". The statement would be misleading. A diSPIM provides isotropic spatial resolution at high speed.

P. 7: The authors stated "These data are consistent with increased microtubule-based movement in the absence of the actin cytoskeletal network". Please add a proper reference to support this statement or rewrite the sentence.

Conclusion:

I can't comment on the originality or significance of the study, as I am not familiar with the current field of virology.

In general, the paper is clearly written. As I have pointed out above, there are significant shortcomings with respect to a thorough consideration of the weaknesses of the imaging method. These could be addressed by the additional data as mentioned here.

Reviewers' comments:

Reviewer #1 (Remarks to the Author):

Influenza A viruses (IAVs) have a segmented RNA genome which is replicated in the nucleus of infected cells. It is poorly understood how the newly synthesized viral RNA segments (vRNAs), in the form of viral ribonucleoproteins (vRNPs), are transported from the nucleus to the plasma membrane and how they assemble into bundles of 8 different vRNA segments to be incorporated into budding progeny virions. There is evidence for a role of the Rab11 GTPase in these processes, but how Rab11 is mechanistically involved remains unclear. This study by Bhagwat et al. expands upon previous work from the same team, which aimed to investigate the role of microtubules in influenza viral RNA (vRNA) transport and assembly (Nturibi et al, Journal of Virology 2017). Their initial study relied on immunofluorescence colocalization analysis and fluorescence in situ hybridization on fixed cells. Here the authors used high spatiotemporal resolution light-sheet microscopy to characterize Rab11A motion in live cells during infection with an IAV or a Respiratory Syncytial virus (RSV), as well as influenza vRNP movements, in the presence or absence of drugs that specifically disrupt the microtubule or actin cytoskeleton filaments. Based on the obtained images, the authors conclude that i) Rab11A motion is primarily dependent on microtubules and not actin in uninfected cells, ii) infection with IAVs slows down Rab11A motion and promotes more stalled motion, whereas infection with RSV alters Rab11A motion in the opposite way iii) influenza vRNPs motion is essentially independent from microtubules.

Overall this is a very well-presented study, in which the imaging data support the conclusions drawn by the authors. The workflow is well explained and convincing. The quality of the live imaging data is impressive. This study contributes to advance the understanding of influenza virus vRNA intracellular transport. However, the advance would be much more significant if additional experiments were performed to provide more mechanistic insights.

(1) The presented data raise the possibility that vRNP transport is independent of the bright GFP-Rab11A foci, which does not necessarily mean “independent of Rab11A”, as discussed by the authors. However evidence is indirect: it relies mostly on the observation that upon nocodazole treatment, the motion of GFP-Rab11A foci is decreased, the colocalization of PAVRNA and Rab11A decreases, whereas the motion of PA::GFP puncta remains unaltered. However, as shown in the Supplementary Figure 3C, about 40% of total PAVRNA and Rab11A spots show colocalization in untreated cells.

Evidence would be much stronger if the tracking of Rab11A and PA was performed simultaneously in live cells. Here the authors used alternatively a GFP-Rab11 cell line and a PA::GFP virus. Is there any technical limitation to the use of a two-colour experimental setting ?

We thank reviewer 1 for their thoughtful insights. Two-color imaging has been historically difficult because it has been technically challenging to rescue a virus with PA

tagged to a red fluorescent protein. We were recently able to overcome this hurdle and have generated a WSN PA::mRuby virus. Using this new fluorescent tool we have performed two-color imaging in A549 Rab11A GFP cells infected with PA::mRuby (Figure 6, movie S11, and supplementary Figure 12 in the revised manuscript). In these cells we were able to separate the Rab11A-GFP spots into those that colocalized with PA::mRuby and those that did not. Analysis of the different GFP-Rab11A populations within a single cell revealed that Rab11A bound by viral RNA (colocalized spots) move slower compared to Rab11A unbound foci. This result is similar to results from our single-color imaging datasets. A new section detailing these additional studies has been added to the revised manuscript (page 12). One limitation to note is that the two-color imaging is captured at a different temporal rate (interval between successive timepoints = 2.8 s) than the single-color datasets (interval = 0.7 s), making it difficult to directly compare values such as straightness or arrest coefficient, since the lag between frames is much larger in the two-color system. This 4x slower acquisition rate is due to a larger gap between acquisitions needed to reduce background signal in our current device. Further modifications to our system should resolve this time lag, but we are not set up yet to make those modifications. We include this caveat in the manuscript on page 13 lines 29-37.

However, even with the time delay, our imaging is still faster than any previous imaging of Rab11A foci, and fast enough to examine the transport dynamics of PA::mRuby that is colocalized with GFP-Rab11A. About 50% of the PA::mRuby spots are not colocalized with GFP-Rab11 and they move faster compared to the colocalized spots.

(2) Previously published work suggest that KIF13A, a member of the kinesin-3 family, mediates influenza vRNP transport (Ramos-Nascimento et al, J Cell Sci 2017) and that vRNPs compete with Rab11 FIP effectors (Vale-Costa et al, J Cell Sci 2016). Investigating the effect of KIF13A depletion/overexpression and FIP overexpression on Rab11A and vRNP motion would provide further mechanistic information.

To address this comment we performed many studies examining KIF13A and Rab11A in A549 cells with or without viral infection. First, we wanted to assess whether KIF13A was associated with Rab11A in A549 cells, so we performed an IP mass spec and pull-down assay but found no KIF13A associated with Rab11A in infected or uninfected cells. However, we did find KIF13A present in A549 cells. Other KIF and Rab11A FIP protein members were associated with Rab11A in our mass spectrometry assay so we are confident in our pull-downs. However, addressing the larger concern about the mechanism of influenza viral RNA transport on Rab11A endosomes, we found a decrease in the amount of dynein bound to Rab11A during influenza infection by mass spectroscopy. We confirmed this with immunoprecipitation in A549 Rab11A-GFP cells. This observation supports our model that the stoichiometry of the motors is disrupted during viral infection. We have added the mass spec data (Figure 7a, b and c), negative KIF13A data (Supplemental Figure S14), and validated IP data (Figure 7d) to this revised manuscript. In addition, we have added discussion comparing our results to those observed in the Ramos-Nascimento et al, J Cell Sci 2017, Vale-Costa et al, J Cell

Sci 2016 and Vale-Costa et al. *Small GTPases*, 2017 *papers, please see lines page 15 lines 36-42.*

Other comments.

(3) Figure 1 : the workflow is well explained, as well as how the CF50 values will be compared to evaluate the effect of different experimental conditions. Consistently, the legend of Supplementary Figures 2, 6 and 8 refer to the CF50 : « Each data point represents an individual analyzed cell using CF50 ». For some reason, it is not the case in the legends of the main Figures 3, 4, 5 and 6. They refer to « mean speed » « mean velocity », which is confusing.

We have added clarification to the figure legends regarding the use of CF50 in figures 3, 4, 5 and 6.

(4) In Figure 2 and Supplementary Figure 5, the size of « GFP-Rab11 foci » and « PA::GFP puncta » is in the same range (0.5 – 1 µm in diameter). In the discussion the authors raise the possibility that the bright GFP-Rab11 foci may represent Rab11 associated with recycling endosomes. Would this imply that the PA ::GFP puncta may represent groups of vRNPs ? and/or vRNPs associated with transport vesicles ? this point should be discussed.

We have added information regarding the characterization of our custom-built diSPIM, and confirm that the resolution is ~300 nm in x, y, and z. We discuss that within a diffraction limited spot there are very likely more than one vRNP and Rab11A foci. Please refer to lines Materials and Methods (page 30, lines 32-34).

(5) Supplementary Figure 3B and page 19. The statement « H1N1pdm PA ::GFP virus showed robust growth in A549 cells... » is inaccurate, as the PA ::GFP virus grows to very low titers compared to its wild-type virus counterpart.

We have modified this statement on page 27, lines 41-43, to read “H1N1pdm PA::GFP virus showed growth in A549 cells in the presence of nocodazole and latrunculin A that was indistinguishable from DMSO (see Figure S6b)”.

(6) Supplementary Figure 3C. The numbers of counted « PA vRNA » and « Rab11A » spots should be indicated.

We have included these values in supplementary Figures S3, S5, S9, S11 and S12.

(7) When discussing the opposite effect of IAV and RSV infection on Rab11A motion, the authors should not exclusively refer to the segmented versus non-segmented nature of their genome, and be more comprehensive.

Additional details of viral lifecycle of IAV and RSV, specifically the site of replication, has been added to the discussion. Please see page 11 lines 13-14 and lines 23-25.

(8) page 7. « These data are consistent with increased microtubule-based movement in the absence of the actin cytoskeletal network ». It is unclear whether this a statement based on available literature (in this case the appropriate references should be added) or a speculation.

We have added a reference for this claim (Howard, J. Mechanics of Motor Proteins and the Cytoskeleton. Chapter 13, Table 1, pp. 214-215 (Sinauer Associates, 2001). We appreciate the reviewer drawing our attention to this oversight.

(9) WSN/1933 or A/WSN/1933 should be spelled A/WSN/33.

We have adjusted this.

(10) The references 1 and 2 are inappropriately numbered.

We have corrected this.

Reviewer #2 (Remarks to the Author):

OVERVIEW:

Several viruses have been shown to use and manipulate Rab11 during their lifecycles (as reviewed in (Bruce et al. 2012)). In this manuscript, the authors have used light sheet microscopy approaches to generate 3D images of the cell and characterize what happens to Rab11 in the presence of cytoskeleton acting drugs and infection with influenza A and B viruses (IAV, IBV, respectively) as well as with respiratory syncytial virus (RSV). The manuscript confirms that RSV increases the movement of Rab11 whilst IAV leads to stalling of Rab11 movement (Amorim et al. 2011; Avilov et al. 2012a; Avilov et al. 2012b; Chambers & Takimoto 2010; Eisfeld et al. 2011; Kawaguchi et al. 2015a; Momose et al. 2011; Utley et al. 2008; Vale-Costa et al. 2016). Similarly to others, the authors hypothesize that the differential modulation of Rab11 during infection relates to the nature of their viral genomes being non-segmented (RSV) or segmented (IAV, IBV), but do not prove this assumption. Conversely to having to transport one segment for incorporation in a virion as RSV, IAV and IBV require that the eight different segments, that constitute the viral genome, form a supra-molecular complex to assemble fully infectious particles. The manuscript presents an interesting methodology to quantify and statistically analyze vesicular trafficking but it does not represent a significant advancement in the knowledge of how Rab11 is manipulated during the different viral infections.

MAJOR ISSUES:

- The study here presented does not provide compelling mechanistic and conceptual advance. A number of previous studies have indicated that IAV viral ribonucleoproteins

(vRNPs) associate with Rab11A after their delivery to the cytoplasm in late-stage infection (Amorim et al. 2011; Avilov et al. 2012a; Avilov et al. 2012b; Momose et al. 2011). It was also repeatedly shown that infection altered Rab11A in a number of ways including cellular distribution (from small to big foci), binding to FIPs and to microtubules (Amorim et al. 2011; Avilov et al. 2012a; Avilov et al. 2012b; Einfeld et al. 2011; Kawaguchi et al. 2015a; Momose et al. 2011; Vale-Costa et al. 2016). Recent manuscripts have, however reinforced that RNPs use a Rab11-independent transport, binding to the endoplasmic reticulum (de Castro Martin et al. 2017) or as this author has shown, having uncoupled subcellular locations in the presence of nocodazole (Nturibi et al. 2017). The stalling of Rab11 movement during influenza A infection has also been shown, namely competition between FIPs and RNPs for Rab11 binding (Kawaguchi et al. 2015b; Vale-Costa et al. 2016). All the data has led to the proposal that the virus creates a platform to concentrate the eight different RNPs to promote the formation of supramolecular complexes. In relation to paramyxoviruses, it was reported that Sendai virus and Respiratory syncytial virus use Rab11. In the case of RSV however, it was demonstrated that the virus relied on FIP2 and binding of the genome did not compete with FIP adaptor. Therefore, the Bhagwat paper confirms previously reported data, but does not add mechanistic insight.

We respectfully disagree with Reviewer 2 that we do not provide any mechanistic insight. In this study, we are the first to examine the movement of Rab11A during a productive infection with high spatiotemporal resolution and track the movement of Rab11A and vRNP in three dimensions. Previous work referenced by Reviewer 2 on live cell imaging of Rab11A was performed in 293T cells and was not quantitatively measured nor was it taken at a temporal resolution suitable for accurate transport analysis (Amorim et al JV 2011). All of the other studies referenced by this Reviewer used static imaging analysis of Rab11A during viral infection. Therefore, we provide a critical and quantitative assessment of Rab11A dynamics during multiple viral infections in a relevant cell line. With our unique methodology we demonstrate that Rab11A movement is decreased during influenza viral infection and suffers more arrests, while it is sped up during RSV infection and travels over a much more direct path compared to uninfected and influenza infected cells. These observations in themselves are novel and not confirmation of previous reports, as suggested by the reviewer.

However, to enhance the mechanistic insight into the manipulation of Rab11A by influenza virus, we performed mass spectrometry of Rab11A-GFP interacting proteins in A549 cells either uninfected or infected with 2009 H1N1 pandemic virus; please see the new Figure 7a,b and c. Comparison of Rab11A-GFP proteins in uninfected and infected cells revealed decreases in the level of dynein, which was confirmed by immunoprecipitation followed by western blot (Figure 7d, Supplementary figure S14). Our mass spec data provide a novel mechanistic finding that the levels of dynein motors are reduced on Rab11A vesicles during viral infection and this leads to altered Rab11A vesicle movement. Therefore, we significantly extend previous observations made primarily from static images with lab-adapted strains of influenza viruses, to describe a mechanism by which Rab11A movement is altered in influenza infected cells.

- All the graphs show that uninfected and untreated cells have a wide dispersion of the mean values calculated. 10 cells were analyzed for this condition. However, for all other treatments the number of the cells analyzed was lower, varying from 4-6 cells. The case of Latrunculin A treatment for example of Fig 2, the cells have a bimodal behavior, with some being affected by the treatment and others not. Analyzing the same number of cells is preferential, especially considering the dispersion of untreated samples.

We analyzed thousands of tracks per cell, which were distilled down to one value, so although the result may appear as if it is derived from only a few data points, these actually represent many thousands of tracks. We find the data to be very robust and consistent between cells.

- The number of moving Rab11/RNPs is very different between conditions. Bigger aggregates are brighter than the smaller versions, which will bias the analyses and acquisition of moving targets. This is particular clear when analyzing S6-S8. Please comment. The inclusion of the number of moving vesicles would complement the data. Also, actin/microtubule-based movements have very different behaviors, and different cellular localization. The methodology described would allow discriminating movements relatively to their cellular location - close to the nucleus or to the plasma membrane, which would be an insightful way to look at Rab11 positioning during viral infection, showing novel and relevant alterations to this molecule for IA, IB and RSV.

Data in the original S6 and S8 figures were tracking influenza vRNP not Rab11A-GFP. In our system we have not observed 'aggregates' of PA-GFP or Rab11A-GFP in live cells. Also our tracking algorithms include all spot intensities in the tracking analysis. Supplemental figure S15 displays violin plots of the Rab11A foci intensity in tracks from all cells of various conditions. These data conclusively demonstrate similar signal intensities in Rab11A untreated, nocodazole treated, latrunculin treated, uninfected and infected cells. Therefore, we do not believe that there is a bias in the tracking based on spot intensities. We do agree with the Reviewer that examination of Rab11A movement in relation to cellular features would be powerful. However, three-dimensional intracellular tracking of spots is a difficult problem and we are working closely with cell biologists to develop additional algorithms to answer these questions. At this time, we feel that they are beyond the scope of this study.

- Videos for direct comparison are very different. This includes the set Supplementary movie 6 to Supplementary movie 8, in which Sup movie8 is formatted in a very distinct manner and does not allow direct comparison.

Based on this comment, we have modified movie S8 such that it shows the same dataset as before, but in the same orientation as movie S6 and S7.

- Representative Movies for Figure 6 have not been included.

Based on this comment, we have now included additional movies for figure 5 (previously called figure 6, please see movie S9 and movie S10).

REFERENCES

- Amorim, M. J., Bruce, E. A., Read, E. K., Foeglein, A., Mahen, R., Stuart, A. D., & Digard, P. 2011. A Rab11- and microtubule-dependent mechanism for cytoplasmic transport of influenza A virus viral RNA. *J Virol*, 85(9): 4143-4156.
- Avilov, S. V., Moisy, D., Munier, S., Schraidt, O., Naffakh, N., & Cusack, S. 2012a. Replication-competent influenza A virus that encodes a split-green fluorescent protein-tagged PB2 polymerase subunit allows live-cell imaging of the virus life cycle. *J Virol*, 86(3): 1433-1448.
- Avilov, S. V., Moisy, D., Naffakh, N., & Cusack, S. 2012b. Influenza A virus progeny vRNP trafficking in live infected cells studied with the virus-encoded fluorescently tagged PB2 protein. *Vaccine*, 30(51): 7411-7417.
- Bruce, E. A., Stuart, A., McCaffrey, M. W., & Digard, P. 2012. Role of the Rab11 pathway in negative-strand virus assembly. *Biochem Soc Trans*, 40(6): 1409-1415.
- Chambers, R., & Takimoto, T. 2010. Trafficking of Sendai virus nucleocapsids is mediated by intracellular vesicles. *PLoS One*, 5(6): e10994.
- de Castro Martin, I. F., Fournier, G., Sachse, M., Pizarro-Cerda, J., Risco, C., & Naffakh, N. 2017. Influenza virus genome reaches the plasma membrane via a modified endoplasmic reticulum and Rab11-dependent vesicles. *Nat Commun*, 8(1): 1396.
- Eisfeld, A. J., Kawakami, E., Watanabe, T., Neumann, G., & Kawaoka, Y. 2011. RAB11A is essential for transport of the influenza virus genome to the plasma membrane. *J Virol*, 85(13): 6117-6126.
- Kawaguchi, A., Asaka, M. N., Matsumoto, K., & Nagata, K. 2015a. Centrosome maturation requires YB-1 to regulate dynamic instability of microtubules for nucleus reassembly. *Sci Rep*, 5: 8768.
- Kawaguchi, A., Hirohama, M., Harada, Y., Osari, S., & Nagata, K. 2015b. Influenza Virus Induces Cholesterol-Enriched Endocytic Recycling Compartments for Budozone Formation via Cell Cycle-Independent Centrosome Maturation. *PLoS Pathog*, 11(11): e1005284.
- Momose, F., Sekimoto, T., Ohkura, T., Jo, S., Kawaguchi, A., Nagata, K., & Morikawa, Y. 2011. Apical transport of influenza A virus ribonucleoprotein requires Rab11-positive recycling endosome. *PLoS One*, 6(6): e21123.
- Nturibi, E., Bhagwat, A. R., Coburn, S., Myerburg, M. M., & Lakdawala, S. S. 2017.

Intracellular Colocalization of Influenza Viral RNA and Rab11A Is Dependent upon Microtubule Filaments. *J Virol*, 91(19).

Utley, T. J., Ducharme, N. A., Varthakavi, V., Shepherd, B. E., Santangelo, P. J., Lindquist, M. E., Goldenring, J. R., & Crowe, J. E., Jr. 2008. Respiratory syncytial virus uses a Vps4-independent budding mechanism controlled by Rab11-FIP2. *Proc Natl Acad Sci U S A*, 105(29): 10209-10214.

Vale-Costa, S., Alenquer, M., Sousa, A. L., Kellen, B., Ramalho, J., Tranfield, E. M., & Amorim, M. J. 2016. Influenza A virus ribonucleoproteins modulate host recycling by competing with Rab11 effectors. *J Cell Sci*, 129(8): 1697-1710.

Reviewer 3 response:

1. The authors provide very little detail about the equipment and performance information for their diSPIM system. For example, which camera, lasers and objective lenses were used in the measurements? What's the actual spatial resolution? Does the system really provide an isotropic spatial resolution? How is the PSF across the FOV in different z positions? What's the light sheet thickness? What's the beam's power density at the sample plane? Because the imaging experiments were performed on a 'home-built' microscope in Dr. Lakdawala's lab, they should verify the system performance and describe the information elsewhere besides the reference [52].

In the "Light-sheet microscopy and image analysis" section of Materials and Methods, we have added the following information:

- a) details about the diSPIM hardware (including the objectives, lasers, cameras, dichroics, acousto-optical tunable filter)*
- b) characterization of imaging resolution along the x, y and z axes using 100-nm fluorescent beads that includes full-width at half-maximum (FWHM) before and after deconvolution and fusion (please see Figure S1 and Table S2).*
- c) light-sheet widths characterized using fluorescent spheres*
- d) laser powers used for imaging for both 488-nm and 561-nm lasers*

2. To characterize the movement of fluorescent foci would be affected by sample preparation. GFP-tagged proteins were over-expressed and imaged in live. How does the over-expression of the proteins affect the cellular physiology? How does the presence of the GFP-tag affect the movement? The questions could have been answered by appropriate control experiments in which proteins with known movement were imaged and quantified in the same condition.

We characterized the effect of overexpression and stable transfection of GFP-Rab11A on A549 cells by performing live imaging of the uptake and recycling of Alexa 568 dye-tagged transferrin. GFP-Rab11A and native A549 cells were incubated with 25 µg/mL of labeled transferrin in imaging media (Fluorobrite DMEM, Gibco) with 1% bovine serum albumin (BSA) according to manufacturer protocols. The transferrin solution was replaced with imaging media and the samples were immediately imaged. The datasets were then subjected to the same image processing and analysis as our other datasets

and the CF50 for all motion parameters were compared for multiple cells. This graph has now been added to the supplementary information as Supplementary Figure S3. We have observed a small but statistically significant difference in the speed of transferrin puncta, with GFP-Rab11A movement being slower. No differences were found to be statistically significant in any of the other parameters.

In all our datasets, we directly compare the effect of a treatment (drugs or infection) within the same type of cell, and no comparison is made between the treatment for A549 cell vs the same treatment for a GFP-Rab11A cell. Thus, due to the combination of the small effect and direct comparison within the same cell type, we believe that our results and conclusions are still rigorous. We agree that it is important to note the effect of stable transfection on the cells and have accordingly added the following sentences to the section **“Rab11A motion is dependent upon microtubules and not actin in A549 cells”**: We compared the transport and recycling of Alexa-568-tagged transferrin in GFP-Rab11A cells against unmodified A549 cells and found them to be similar by most measures except mean speeds, which showed a small, albeit, statistically significant difference (see Figure S3). Thus, in this report, we do not directly compare the results obtained in unmodified A549 cells to those obtained in GFP-Rab11A-expressing A549 cells.

3. Vesicle tracking not only requires high-speed imaging but also a rigorous tracking algorithm. The authors used Imaris to identify locations of fluorescent foci. When looking at Figure 2 and S5, I wonder if the “Spots” function is rigorous enough. It’s because some of GFP foci were unidentified or not donated by red spots. A couple of questions also arise, “Can the authors reliably identify GFP foci when they are packed?” “How is the identification affected by foci density?”

The reviewer brings up an important point about spot identification and rigor. Accordingly, we have added a brief discussion in the Materials and Methods section on Light Sheet Microscopy on how we account for as many real spots as possible without counting spurious spots arising from thermal or dark current noise in the camera. Please see page 30, lines 28-35: “Briefly, spot generation was achieved manually for each dataset using intensity at the center of each spot as the threshold. The thresholds was lowered till each moving signal density was assigned a spot. The spots were tracked over time to generate motion statistics for each cell. Based on the resolution measurement, we cannot distinguish individual fluorescent emitters within a diffraction limited sphere of diameter ~300 nm. Thus each identified spot in our dataset may contain multiple GFP molecules, and hence multiple Rab11A proteins and multiple IAV vRNP segments. By filtering out tracks smaller than 3 steps, we can eliminate counting of spurious spots that arise due to thermal or dark current noise in the camera.”

We have also referenced in the manuscript step-by-step protocols that we have previously published for imaging with the diSPIM, where the spot identification and tracking is discussed in greater length [Bhagwat, A. R., Le Sage V & Lakdawala, S. S. in Influenza Virus: Methods and Protocols Vol. 1836 (ed Y. Yamauchi) (Humana Press, 2018)].

For Figure 1, we have revised the figure caption to be clearer about what is being shown : “Fluorescent puncta are identified and tracked over time in Imaris (green spheres show GFP-Rab11A spot centers from a single time frame corresponding to tracks (red lines, only tracks with displacement > 2 μm shown for clarity) over all time frames).”

In Figure 2, for clarity only those spots that are involved in the sequence of splitting and merging have been marked using a red sphere. Some previous frames (timepoint $t=48.5$ and 49.3) were omitted for concise representation. The one sphere on the bottom left that is not part of the yellow track split in the previous frame ($t=49.3$ s) and hence is still displayed. We have modified the caption to read: “Yellow track shows location history and current direction of motion. For clarity, only centers of those GFP-Rab11A foci that take part in this interaction are denoted by red spots.”

In Figure S8 (previously S5), it is difficult for all spots to be identified perfectly for a number of reasons – 1) the spots may be spurious blips that go in and out of existence, 2) blinking fluorophores make it difficult for very dim spots to be correctly identified and counted as a continuous track, or 3) the spots are stable, but so dim that there is not enough intensity gradient for the spot identification algorithm to accurately identify the spot. Such spots will not be accounted for until better algorithms are discovered. The algorithm built into Imaris is the best one we have identified so far that will accurately track spots in all 3 dimensions. All other algorithms (TrackMate, UTrack, trackPy etc.) only work well in 2 dimensions. There is a 3 dimensional algorithm being developed by Jaqaman et al. (inventors of u-track) but is not publicly available yet.

A high foci density is definitely more of a problem for tracking than spot identification. We check spot identification by two methods: 1) assign spots and play the dataset movie over time to check if all moving signal densities are being accounted 2) The datasets at any given time are rotated along multiple axes to check parallax, so that the entire 3D region of spots can be sensed by human vision and all signal densities assigned spots. Using these techniques, we feel confident that all spots are accurately identified even in dense fields.

4. An artifact is observed in Movie S3, S4, S5 and S8. The shapes of GFP foci in the XZ plane are diagonally elongated (see the attached pictures). It's probably because the raw data are not so good or image registration is not optimal. With such a condition, it is difficult to evaluate the accuracy of their data.

Usually *ab initio* 3-D registration is performed on the first dataset, and subsequent datasets are registered by starting with the value of the first dataset. We checked whether this was the origin of the diagonal elongation by registering each dataset independently, but this did not change the outcome. There is also no difference between using our custom CUDA/GPU based code vs. the previously published plugin (Nat Protoc. 2014 Nov; 9(11): 2555–2573.) for the free software MIPAV (NIH). During characterization of the diSPIM, we came across agglomerations of multiple 100-nm beads that form bright foci. During registration and deconvolution of the bright foci, we observed a slight, but similar cross pattern. See image included below (A and B are pre-registration and deconvolution, whereas C and D are post-deconvolution and

registration). Red circles show the resulting cross pattern. This is not observed to be significant for images of single spheres (Figure S1).

We observe that in spite of the cross pattern, the identification of the center of the spot is unambiguous, and compared to the pre-registration image, the post-registration and deconvolution image has vastly improved localization (from $1.4 \pm 0.2 \mu\text{m}$ to $0.35 \pm 0.03 \mu\text{m}$, see Figure S1 and Table S2).

Compared to stationary 100-nm beads, fast moving vesicles can exhibit a small change in position between the acquisition of the two views. This change in position is random at any point in the cellular volume and the registration algorithm finds the best overall registration for the volume pair. In this process, it is possible that some spots will undergo smearing. Despite the smearing, the center of signal density post fusion is easily identified and we believe that the overall statistics of the fast motion are accurate. We are in the process of implementing a co-temporal image acquisition of both views, which will ameliorate the problem of fusing moving spots, but for now this limitation persists.

Specific comments:

P.2: The authors stated "... (diSPIM) was developed as a tool for imaging live cells at high spatiotemporal resolution...". The statement would be misleading. A diSPIM provides isotropic spatial resolution at high speed.

We have rewritten the statement: "Recently, dual-view inverted selective plane illumination microscopy (diSPIM) was demonstrated as a tool useful for imaging live cells at high temporal resolution (200 Hz frame rate, 2 volumes/s) with isotropic resolution (~330 nm) over long time intervals (>12 h) without imparting the phototoxicity or bleaching inherent to spinning-disk confocal microscopes".

P. 7: The authors stated “These data are consistent with increased microtubule-based movement in the absence of the actin cytoskeletal network”. Please add a proper reference to support this statement or rewrite the sentence.

We have now added a reference (Howard, J. Mechanics of Motor Proteins and the Cytoskeleton. Chapter 13, Table 1, pp. 214-215 (Sinauer Associates, 2001)) that compares the speed, motility, force and other relevant characteristics of multiple kinesins, dyneins and myosins. The sentence has been rewritten for clarity: “These data (especially the increased speed) are consistent with increased microtubule-based movement of vesicles using kinesins and dyneins in the absence of the actin cytoskeletal network”

Reviewers' comments:

Reviewer #1 (Remarks to the Author):

The new data (two-color imaging and mass spectrometry analysis of Rab11-associated proteins in IAV-infected versus mock-infected cells) significantly strengthen the paper. The authors' conclusions are supported by the data. The technical challenges and the questions that remain open are clearly discussed.

This is a very interesting study that sheds light on influenza vRNP trafficking and provides a framework for further studies in the field.

I am happy with the revised version and have no further comments.

Reviewer #3 (Remarks to the Author):

By addressing all comments that this reviewer pointed out, the authors made a substantial improvement in the manuscript. The imaging approach they developed for tracking particles meets the needs of cell biologists. I now believe that the observation in this manuscript is of interest to this journal.

While the technology employed in the present manuscript is outstanding, and allows analysing influenza RNPs' transport to the plasma membrane with great detail and resolution, the conclusions are, for the first part, not novel and, for the second part, rely largely on correlations. The message conveyed in the 1st version of the manuscript was that influenza A virus RNPs used Rab11 vesicles and that during infection Rab11 movement decreased. Rab11-RNP movement relied mostly on microtubules, although actin was also involved, and in addition RNPs could also be transported independently of Rab11. This was the understanding in the field before this work and was not further dissected by it. As such, the increment in knowledge was considered by this reviewer modest at best. The subsidiary aim, which is on Rab11 usage by other viruses, including of RSV, was also known. The novel data is on influenza B virus, although given the similarities between influenza A and B RNPs the results are expected, as the authors state. These however, should be explored in more detail.

In the revised manuscript, the authors revisited the mechanism proposed for arresting Rab11 vesicular movement upon influenza infection. The authors argue for the arrest of Rab11 movement during influenza infection being caused by lack of dynein and for a direct interaction between Rab11 vesicles and dynein. I think that the work has potential but that at the moment is descriptive, with important conclusions being overinterpretations obtained from correlations without demonstration of causation and need substantial validation. I have two major criticisms in relation to this. First, the authors identify a reduction in dynein as being the cause of Rab11 vesicular arrest in infection, but do not provide evidence that the reduction in Rab11-dynein could affect Rab11 movement. Second, the authors state that dynein binding to Rab11 occurs independently of FIPs, without proving that this is actually the case. Both demonstrations are essential to argue for a model in which the loss of Rab11-dynein direct binding originates arrest in Rab11 movement during IAV infection.

Also, it was disappointing that the authors did not address the concerns expressed in the 1st round of review. In particular, relating to the importance of reproducibility, it had been commented that findings relying

on the analysis of only four - five immortalized cells (full of mutations) are not up to current standards. I address this, as well as other comments, in more detail below.

For simplicity, all the previous interactions are included and highlighted in grey. In black are the 1st comments of this reviewer. The second round of answers is in green, written below each of the 1st answers provided by the authors (in blue).

OVERVIEW:

Several viruses have been shown to use and manipulate Rab11 during their lifecycles (as reviewed in (Bruce et al. 2012)). In this manuscript, the authors have used light sheet microscopy approaches to generate 3D images of the cell and characterize what happens to Rab11 in the presence of cytoskeleton acting drugs and infection with influenza A and B viruses (IAV, IBV, respectively) as well as with respiratory syncytial virus (RSV). The manuscript confirms that RSV increases the movement of Rab11 whilst IAV leads to stalling of Rab11 movement (Amorim et al. 2011; Avilov et al. 2012a; Avilov et al. 2012b; Chambers & Takimoto 2010; Einfeld et al. 2011; Kawaguchi et al. 2015a; Momose et al. 2011; Utley et al. 2008; Vale-Costa et al. 2016). Similarly to others, the authors hypothesize that the differential modulation of Rab11 during infection relates to the nature of their viral genomes being non-segmented (RSV) or segmented (IAV, IBV), but do not prove this assumption. Conversely to having to transport one segment for incorporation in a virion as RSV, IAV and IBV require that the eight different segments, that constitute the viral genome, form a supra-molecular complex to assemble fully infectious particles. The manuscript presents an interesting methodology to quantify and statistically analyze vesicular trafficking but it does not represent a significant advancement in the knowledge of how Rab11 is manipulated during the different viral infections.

MAJOR ISSUES:-The study here presented does not provide compelling mechanistic and conceptual advance. A number of previous studies have indicated that IAV viral

ribonucleoproteins(vRNPs) associate with Rab11A after their delivery to the cytoplasm in late-stage infection (Amorim et al. 2011; Avilov et al. 2012a; Avilov et al. 2012b; Momose et al. 2011). It was also repeatedly shown that infection altered Rab11A in a number of ways including cellular distribution (from small to big foci), binding to FIPs and to microtubules (Amorim et al. 2011; Avilov et al. 2012a; Avilov et al. 2012b; Eisfeld et al. 2011; Kawaguchi et al. 2015a; Momose et al. 2011; Vale-Costa et al. 2016). Recent manuscripts have, however reinforced that RNPs use a Rab11-independent transport, binding to the endoplasmic reticulum (de Castro Martin et al. 2017) or as this author has shown, having uncoupled subcellular locations in the presence of nocodazole (Nturibi et al. 2017). The stalling of Rab11 movement during influenza A infection has also been shown, namely competition between FIPs and RNPs for Rab11 binding (Kawaguchi et al. 2015b; Vale-Costa et al. 2016). All the data has led to the proposal that the virus creates a platform to concentrate the eight different RNPs to promote the formation of supramolecular complexes. In relation to paramyxoviruses, it was reported that Sendai virus and Respiratory syncytial virus use Rab11. In the case of RSV however, it was demonstrated that the virus relied on FIP2 and binding of the genome did not compete with FIP adaptor. Therefore, the Bhagwat paper confirms previously reported data, but does not add mechanistic insight.

We respectfully disagree with Reviewer 2 that we do not provide any mechanistic insight. In this study, we are the first to examine the movement of Rab11A during a productive infection with high spatiotemporal resolution and track the movement of Rab11A and vRNP in three dimensions. Previous work referenced by Reviewer 2 on live cell imaging of Rab11A was performed in 293T cells and was not quantitatively measured nor was it taken at a temporal resolution suitable for accurate transport analysis (Amorim et al JV 2011). All of the other studies referenced by this Reviewer used static imaging analysis of Rab11A during viral infection. Therefore, we provide a critical and quantitative assessment of Rab11A dynamics during multiple viral infections in a relevant cell line. With our unique

methodology we demonstrate that Rab11A movement is decreased during influenza viral infection and suffers more arrests, while it is sped up during RSV infection and travels over a much more direct path compared to uninfected and influenza infected cells. These observations in themselves are novel and not confirmation of previous reports, as suggested by the reviewer.

Contrary to what is being suggested by the authors, the diversity in the systems used - live to static, variety of cell lines (A549s, Vero, 293Ts and MDCKs) and viruses - is a strength in the proposed manipulation of Rab11 pathway during infection with influenza and members of the paramyxoviridae family and should not be considered a weakness. This is particularly relevant when the findings in the present manuscript corroborate the findings of previous articles regarding RNP movement, dependency on Rab11 and on the cytoskeleton. The authors should read and cite the manuscript Avilov et al, JVI, 2012, 86, 1433. This manuscript quantified the migration lengths, and instant velocities of RNPs, taking into account Rab11, in productive infections (using split GFP system) in relevant lines (including A549s) treated with nocodazole, cytochalasin D or both. They reported a Rab11-dependent, microtubule independent way to transport RNPs. More recently, it was suggested that RNPs could use Rab11 independent means of transportation (Nturibi, et al, 2017, JVI; de castro Martin, 2017, Nature Comms). Details on RSV infection were also reported before. The data on IBV, if more detailed, would be extremely interesting as is novel.

However, to enhance the mechanistic insight into the manipulation of Rab11A by influenza virus, we performed mass spectrometry of Rab11A-GFP interacting proteins in A549 cells either uninfected or infected with 2009 H1N1 pandemic virus; please see the new Figure 7a,b and c. Comparison of Rab11A-GFP proteins in uninfected and infected cells revealed decreases in the level of dynein, which was confirmed by immunoprecipitation followed by western blot (Figure 7d, Supplementary figure S14). Our mass spec data provide a novel mechanistic finding that the levels of dynein motors are reduced on Rab11A vesicles during viral infection and this leads to altered

Rab11A vesicle movement. Therefore, we significantly extend previous observations made primarily from static images with lab-adapted strains of influenza viruses, to describe a mechanism by which Rab11A movement is altered in influenza infected cells.

The pull down followed by mass spectrometry data needs to be appropriately validated. In particular, note that:

- out of the 16 factors with higher changes for Rab11 binding and higher in infected and uninfected cells, five are nuclear proteins. Rab11 has not been reported to shuttle between the nucleus and the cytosol.
- out of the same 16 factors with higher changes for Rab11 binding between infected and uninfected cells, five are viral proteins, of which haemagglutinin was previously demonstrated not to bind Rab11-GTP in infected cells (Momose et al, Plos One, 6m e21123 – Fig 5).

This suggests that unspecific ligations might be occurring in the test tube and underscore the need for validation.

Furthermore, it is important to validate the conclusion that dynein binds Rab11 independent of FIPs. Work in cell biology showed that Rab11-vesicles attach to many different molecular motors via 5 different Rab11 Family-interacting proteins (Chu *et al.*, 2009, *JBC*, 284, 22481-90; Gidon *et al.*, 2012, *Traffic*, 13, 815-33; Horgan *et al.*, 2010, *JCS*, 123, 181-91; Lapierre *et al.*, 2001, *MBC*, 12, 1843-57; Provance *et al.*, 2008, *BMC Cell Biol*, 9, 44; Roland *et al.*, 2011, *PNAS*, 108, 2789-94; Schonteich *et al.*, 2008, *JCS*, 121, 3824-33). In this manuscript, for example, FIP3 and FIP4 have not been detected in cell input or Rab11 pull down assay and therefore two questions should be answered: Are these FIPs not expressed in A549 cells? Can dynein bind Rab11 via FIP2, as demonstrated before (Ducharme et al, 2011, *Cell Log*, 1, 57). Importantly, as the data set for FIPs is incomplete, the authors' claim on data on all FIPs is inaccurate and should be revised (see specific points below).

-All the graphs show that uninfected and untreated cells have a wide dispersion of the mean values calculated. 10 cells were analyzed for this condition. However, for all other treatments the number of the

cells analyzed was lower, varying from 4-6 cells. The case of Latrunculin A treatment for example of Fig 2, the cells have a bimodal behavior, with some being affected by the treatment and others not. Analyzing the same number of cells is preferential, especially considering the dispersion of untreated samples.

We analyzed thousands of tracks per cell, which were distilled down to one value, so although the result may appear as if it is derived from only a few data points, these actually represent many thousands of tracks. We find the data to be very robust and consistent between cells.

Solid conclusions should be derived from truly independent samples. Despite counting thousands of tracks the authors are in reality analysing the behaviour of 4 immortalized cells (carrying many mutations), which is by no means representative of the behaviour of cells or of viruses (carrying many mutations). The error associated with the data of Fig 3-6 supports this view.

- The number of moving Rab11/RNPs is very different between conditions. Bigger aggregates are brighter than the smaller versions, which will bias the analyses and acquisition of moving targets. This is particular clear when analyzing S6-S8. Please comment. The inclusion of the number of moving vesicles would complement the data. Also, actin/microtubule-based movements have very different behaviors, and different cellular localization. The methodology described would allow discriminating movements relatively to their cellular location -close to the nucleus or to the plasma membrane, which would be an insightful way to look at Rab11 positioning during viral infection, showing novel and relevant alterations to this molecule for IA, IB and RSV.

Data in the original S6 and S8 figures were tracking influenza vRNP not Rab11A-GFP. In our system we have not observed 'aggregates' of PA-GFP or Rab11A-GFP in live cells. Also our tracking algorithms include all spot intensities in the tracking analysis. Supplemental figure S15 displays violin plots of the Rab11A foci intensity in tracks from all cells of various conditions. These data conclusively

demonstrate similar signal intensities in Rab11A untreated, nocodazole treated, latrunculin treated, uninfected and infected cells. Therefore, we do not believe that there is a bias in the tracking based on spot intensities. We do agree with the Reviewer that examination of Rab11A movement in relation to cellular features would be powerful. However, three-dimensional intracellular tracking of spots is a difficult problem and we are working closely with cell biologists to develop additional algorithms to answer these questions. At this time, we feel that they are beyond the scope of this study.

This reviewer is not convinced by the violin plots on the similar Rab11 foci intensities for all conditions: the median, higher values and probability differ between treatments. Changes in Rab11 intensity foci have been reported independently by the labs of Kawaoka and Digard and quantitatively assessed in Vale-Costa et al, 2016.

However, as this issue does not affect the main conclusions of the manuscript, and the authors are unable to examine Rab11/RNP movements in relation to cellular structures, this point becomes irrelevant.

ADDITIONAL COMMENTS:

- There are many articles on Myo5B binding to Rab11 via FIP2. To strengthen the data, the authors should validate the uncorrelated behaviour of Myo5B/FIP2 during infection and comment on it.
- In the absence of infection, what is the effect of reducing Rab11-dynein interaction on Rab11 vesicular movement? Also, does the overexpression of dynein interfere with viral infection or with Rab11 movement? As a premise of the direct binding between dynein and Rab11, overexpressing dynein should interfere with Rab11 arrest observed during infection.
- Page 15, lines 30-33

“Recent work using mitochondrial-targeting of FIP proteins suggested that IAV vRNP segments outcompete FIP2 for binding with Rab11A 45. Consistent with this finding we observed a significant decrease in the

association with FIP2 binding to Rab11A during IAV infection (Figure 7b). However, the levels of all other FIPs were not altered”.

Please refer to the work of Momose et al, Plos One, 2011, the first manuscript to analyse FIPs in influenza A virus infected cells. Also, the final sentence needs revision. The mass spectrometry data of both the IP and cell input did not detect FIP3 or FIP4. Therefore, it is inaccurate to state that all other FIPs were not altered because the set of FIPs is incomplete.

Discussion:

- The reduction in dynein is interesting. It has implications to the mechanisms reported on the interactions between molecular motors and Rab11 vesicles, and how the tug of war between different motors contribute to Rab11 movement. The authors should discuss this. In addition, how would a specific reduction in dynein affect Rab11 movement in polarized cells?
- Page 16 Lines 29-36

“Rab11A specifically associates with motor proteins through FIPs, where certain specific FIPs can associate with different motor proteins 17,18,37-39. Recent work using mitochondrial-targeting of FIP proteins found that IAV vRNP segments may outcompete certain FIPs for binding with Rab11A 45, which could lead to a change in the distribution of motors on the Rab11A-containing RE. In this study, we did not identify whether a reduction in dynein was a result of a reduction in the association of the corresponding FIP – Rab11-FIP3 – since only FIP2 was decreased in mass spectrometry studies”.

The work of Momose et al, Plos One, 2011, and the final sentence needs revision for the reasons explained above.

- Page 16 Lines 36-38

In addition, lack of direct interaction between KIF13A and Rab11A in cells suggests that dynein dysregulation and not KIF13A, as previously suggested, may be responsible for disrupting Rab11A intracellular movement during IAV infection.

Please revise your sentence after reanalysing the manuscripts, as you find suitable. What I read from the papers cited is:

The molecular motors proposed to be reduced during influenza infection are the ones operating via FIPs, like dynein (in article 45), but KIF13 does not belong to this group as it binds Rab11 independently of FIPs (in article 46).

- Page 16, Line 42-44

for a microtubule independent mechanism of IAV vRNP transport that may be Rab11A-independent as well.

Please include the paper of Avilov et al, JVI, 2012 and comment on their identification that actin is important for vRNP transport.

Also, the authors use a system overexpressing GFP-Rab11, that although carefully evaluated, is also not the ideal, especially if maintaining endogenous Rab11 in the cell. This means that the percentage of RNPs that bind GFP-Rab11 is underestimated and that the conclusion on “a microtubule independent mechanism of IAV vRNP transport that may be Rab11A-independent as well” is overestimated. Please comment.

- Page 17, Line 18 – 22:

“However, recent work has suggested that vRNP exit the ER in liquid-phase organelles rather than membranous structures⁵¹. Therefore, additional work examining the intracellular movement of individual vRNP segments and ER membranes within a single cell is needed to elucidate the importance of dynamic membranous scaffolds during IAV infection”.

This comment should be revised. The paper Milovanovic, D. et al, 2018, Science (10.1126/science.aat5671) (Pietro di Camilli Lab) shows that vesicles destined for the immunological synapse are concentrated in liquid organelles. Similarly, the liquid organelles described in article 51 contain vesicles in the core and are not mutually exclusive to membranous structures assisting genome assembly.

Point-by-Point Rebuttal.

While the technology employed in the present manuscript is outstanding, and allows analysing influenza RNPs' transport to the plasma membrane with great detail and resolution, the conclusions are, for the first part, not novel and, for the second part, rely largely on correlations. The message conveyed in the 1st version of the manuscript was that influenza A virus RNPs used Rab11 vesicles and that during infection Rab11 movement decreased. Rab11-RNP movement relied mostly on microtubules, although actin was also involved, and in addition RNPs could also be transported independently of Rab11. This was the understanding in the field before this work and was not further dissected by it. As such, the increment in knowledge was considered by this reviewer modest at best. The subsidiary aim, which is on Rab11 usage by other viruses, including of RSV, was also known. The novel data is on influenza B virus, although given the similarities between influenza A and B RNPs the results are expected, as the authors state. These however, should be explored in more detail.

In the revised manuscript, the authors revisited the mechanism proposed for arresting Rab11 vesicular movement upon influenza infection. The authors argue for the arrest of Rab11 movement during influenza infection being caused by lack of dynein and for a direct interaction between Rab11 vesicles and dynein. I think that the work has potential but that at the moment is descriptive, with important conclusions being overinterpretations obtained from correlations without demonstration of causation and need substantial validation. I have two major criticisms in relation to this. First, the authors identify a reduction in dynein as being the cause of Rab11 vesicular arrest in infection, but do not provide evidence that the reduction in Rab11-dynein could affect Rab11 movement. Second, the authors state that dynein binding to Rab11 occurs independently of FIPs, without proving that this is actually the case. Both demonstrations are essential to argue for a model in which the loss of Rab11-dynein direct binding originates arrest in Rab11 movement during IAV infection.

>Author Response August 2019: *To avoid misinterpretation of the text, we have provided a schematic (Figure 8) that presents our model for how influenza viral RNA disrupts dynein association. See also revised text on page 10 lines 22-25.*

Also, it was disappointing that the authors did not address the concerns expressed in the 1st round of review. In particular, relating to the

importance of reproducibility, it had been commented that findings relying on the analysis of only four - five immortalized cells (full of mutations) are not up to current standards. I address this, as well as other comments, in more detail below.

>Author Response August 2019: In the revised text we have increased the number of cells so that all three main conclusions are based on >10,000 tracks from 7-10 cells per condition. This additional analysis did not change any of the conclusions in the manuscript.

For simplicity, all the previous interactions are included and highlighted in grey. In black are the 1st comments of this reviewer. The second round of answers is in green, written below each of the 1st answers provided by the authors (in blue).

OVERVIEW:

Several viruses have been shown to use and manipulate Rab11 during their lifecycles (as reviewed in (Bruce et al. 2012)). In this manuscript, the authors have used light sheet microscopy approaches to generate 3D images of the cell and characterize what happens to Rab11 in the presence of cytoskeleton acting drugs and infection with influenza A and B viruses (IAV, IBV, respectively) as well as with respiratory syncytial virus (RSV). The manuscript confirms that RSV increases the movement of Rab11 whilst IAV leads to stalling of Rab11 movement (Amorim et al. 2011; Avilov et al. 2012a; Avilov et al. 2012b; Chambers & Takimoto 2010; Einfeld et al. 2011; Kawaguchi et al. 2015a; Momose et al. 2011; Utley et al. 2008; Vale-Costa et al. 2016). Similarly to others, the authors hypothesize that the differential modulation of Rab11 during infection relates to the nature of their viral genomes being non-segmented (RSV) or segmented (IAV, IBV), but do not prove this assumption. Conversely to having to transport one segment for incorporation in a virion as RSV, IAV and IBV require that the eight different segments, that constitute the viral genome, form a supra-molecular complex to assemble fully infectious particles. The manuscript presents an interesting methodology to quantify and statistically analyze vesicular trafficking but it does not represent a significant advancement in the knowledge of how Rab11 is manipulated during the different viral infections.

MAJOR ISSUES:-The study here presented does not provide compelling mechanistic and conceptual advance. A number of previous studies have indicated that IAV viral ribonucleoproteins(vRNPs) associate with Rab11A after their delivery to the cytoplasm in late-stage infection (Amorim et al. 2011; Avilov et al. 2012a; Avilov et al. 2012b; Momose et al. 2011). It was also repeatedly shown that infection altered Rab11A in a number of ways including cellular distribution (from small to big foci), binding to FIPs and to microtubules (Amorim et al. 2011; Avilov et al. 2012a; Avilov et al. 2012b; Eisfeld et al. 2011; Kawaguchi et al. 2015a; Momose et al. 2011; Vale-Costa et al. 2016). Recent manuscripts have, however reinforced that RNPs use a Rab11-independent transport, binding to the endoplasmic reticulum (de Castro Martin et al. 2017) or as this author has shown, having uncoupled subcellular locations in the presence of nocodazole (Nturibi et al. 2017). The stalling of Rab11 movement during influenza A infection has also been shown, namely competition between FIPs and RNPs for Rab11 binding (Kawaguchi et al. 2015b; Vale-Costa et al. 2016). All the data has led to the proposal that the virus creates a platform to concentrate the eight different RNPs to promote the formation of supramolecular complexes. In relation to paramyxoviruses, it was reported that Sendai virus and Respiratory syncytial virus use Rab11. In the case of RSV however, it was demonstrated that the virus relied on FIP2 and binding of the genome did not compete with FIP adaptor. Therefore, the Bhagwat paper confirms previously reported data, but does not add mechanistic insight.

We respectfully disagree with Reviewer 2 that we do not provide any mechanistic insight. In this study, we are the first to examine the movement of Rab11A during a productive infection with high spatiotemporal resolution and track the movement of Rab11A and vRNP in three dimensions. Previous work referenced by Reviewer 2 on live cell imaging of Rab11A was performed in 293T cells and was not quantitatively measured nor was it taken at a temporal resolution suitable for accurate transport analysis (Amorim et al JV 2011). All of the other studies referenced by this Reviewer used static imaging analysis of Rab11A during viral infection. Therefore, we provide a critical and quantitative assessment of Rab11A dynamics during

multiple viral infections in a relevant cell line. With our unique methodology we demonstrate that Rab11A movement is decreased during influenza viral infection and suffers more arrests, while it is sped up during RSV infection and travels over a much more direct path compared to uninfected and influenza infected cells. These observations in themselves are novel and not confirmation of previous reports, as suggested by the reviewer.

Contrary to what is being suggested by the authors, the diversity in the systems used - live to static, variety of cell lines (A549s, Vero, 293Ts and MDCKs) and viruses - is a strength in the proposed manipulation of Rab11 pathway during infection with influenza and members of the paramyxoviridae family and should not be considered a weakness. This is particularly relevant when the findings in the present manuscript corroborate the findings of previous articles regarding RNP movement, dependency on Rab11 and on the cytoskeleton. The authors should read and cite the manuscript Avilov et al, JVI, 2012,86, 1433. This manuscript quantified the migration lengths, and instant velocities of RNPs, taking into account Rab11, in productive infections (using split GFP system) in relevant lines (including A549s) treated with nocodazole, cytochalasin D or both. They reported a Rab11-dependent, microtubule independent way to transport RNPs. More recently, it was suggested that RNPs could use Rab11 independent means of transportation (Nturibi, et al, 2017, JVI; de castro Martin, 2017, Nature Comms). Details on RSV infection were also reported before. The data on IBV, if more detailed, would be extremely interesting as is novel.

>Author Response August 2019: *We disagree with the reviewer. No other study cited by the reviewer tracks the dynamics of Rab11A or influenza vRNP in three dimensions. Avilov et al 2012 is cited in the manuscript regarding the analysis of vRNP track length in the absence of microtubules.*

However, to enhance the mechanistic insight into the manipulation of Rab11A by influenza virus, we performed mass spectrometry of Rab11A-GFP interacting proteins in A549 cells either uninfected or infected with 2009 H1N1 pandemic virus; please see the new Figure 7a,b and c. Comparison of Rab11A-GFP proteins in uninfected and infected cells revealed decreases in the level of dynein, which was confirmed by immunoprecipitation followed by western blot (Figure 7d, Supplementary figure S14). Our mass spec data provide a novel mechanistic finding that the levels of dynein motors are reduced on

Rab11A vesicles during viral infection and this leads to altered Rab11A vesicle movement. Therefore, we significantly extend previous observations made primarily from static images with lab adapted strains of influenza viruses, to describe a mechanism by which Rab11A movement is altered in influenza infected cells.

The pull down followed by mass spectrometry data needs to be appropriately validated. In particular, note that:

- out of the 16 factors with higher changes for Rab11 binding and higher in infected and uninfected cells, five are nuclear proteins.

Rab11 has not been reported to shuttle between the nucleus and the cytosol.

>**Author Response August 2019:** *The top RNA binding proteins identified in our mass spectrometry studies, specifically SFPQ, has been shown to bind to PB2 (PMID:25687574) and both SFPQ and NONO are present in influenza virions grown in mammalian cells (PMID:18491320). The presence in purified virions by mass spectroscopy from the Fodor group suggests that these proteins must travel through the cytoplasm for packaging into progeny influenza virions at the plasma membrane. Consistent with this, we have included a figure for the reviewers (below) demonstrating that both SFPQ and NONO are present in the cytoplasm of A549 cells infected with H1N1pdm over time by western blot of fractionated cells. In addition, we demonstrate by immunofluorescence that NONO is cytoplasmic in infected cells and colocalizes with Rab11A GFP. These published and unpublished data are consistent with our interpretation of cytoplasmic interaction between Rab11A and RNA binding proteins. We have modified the text to state this point (page 8 lines 38-39). We are actively pursuing the importance of the RNA binding proteins associated with Rab11A during viral infection and have chosen not to dilute the message this current manuscript, which focuses on the transport dynamics of Rab11A and viral RNA, with our data regarding these RNA-binding proteins.*

[Redacted]

- out of the same 16 factors with higher changes for Rab11 binding between infected and uninfected cells, five are viral proteins, of which haemagglutinin was previously demonstrated not to bind Rab11-GTP in infected cells (Momose et al, Plos One, 6m e21123 - Fig5).

This suggests that unspecific ligations might be occurring in the test tube and underscore the need for validation.

>Author Response August 2019: *HA and Rab11A are both membrane associated proteins and given the high abundance of HA during viral infection it is likely that some HA is present on endosomes leaving the ER or golgi to travel to the plasma membrane.*

Furthermore, it is important to validate the conclusion that dynein binds Rab11 independent of FIPs. Work in cell biology showed that Rab11-vesicles attach to many different molecular motors via 5 different Rab11 Family-interacting proteins (Chu et al., 2009, JBC, 284, 22481-90; Gidon et al., 2012, Traffic, 13, 815-33; Horgan et al., 2010, JCS, 123, 181-91; Lapierre et al., 2001, MBC, 12, 1843-57; Provance et al., 2008, BMC Cell Biol, 9, 44; Roland et al., 2011, PNAS, 108, 2789-94; Schonteich et al., 2008, JCS, 121, 3824-33). In this manuscript, for example, FIP3 and FIP4 have not been detected in cell input or Rab11 pull down assay and therefore two questions should be answered: Are these FIPs not expressed in A549 cells? Can dynein bind Rab11 via FIP2, as demonstrated before (Ducharme et al, 2011, Cell Log, 1, 57). Importantly, as the data set for FIPs is incomplete, the authors claim on data on all FIPs is inaccurate and should be revised (see specific points below).

>Author Response August 2019: *Based on further analysis of proteomic databases, we note that FIP3 and FIP4 do not appear to be expressed in lung cells. Figures are presented here to clarify this point. FIP3 and 4 are detected in a large array of primary tissues and cell lines, as confirmed in ProteomicsDB (Figure 1; PMID: 29106664), the Human Proteome Map (Figure 2; PMID: 24870542), and the Human Protein Atlas (Figure 3; PMID: 16127175; PMID: 25613900). However, despite the massive amount of data in these repositories that originated from highly sensitive experiments, there is essentially no trace of the two proteins detectable in lung cells (highlighted with stars, Figures 1-3). One exception is Rab11 FIP 4 detected in a single cancer lung cell line in ProteomicsDB (HOP-92), which is a different cell line than A549. Moreover, none of the previous studies on Rab11A-RAB11FIP3-RAB11FIP4 interactions were performed in the lung cell line and primarily used over-expression of FIP3 or 4 (Table 1). We have modified the text to address this point (page 8 line 44 and page 9 lines 1-6).*

Figure 1. The expression pattern of Rab11fip3 (A) and Rab11fip4 (B) protein across the human body. Data retrieved from the Proteomics DB database (<https://www.proteomicsdb.org/>)

Table 1. The cell lines used in the studies on Rab11A-RAB11FIP3-RAB11FIP4 protein interaction.

#	PubMed ID	Celltype	Year of publication
1	PMID: 11495908	HeLa; MDCK	2001
2	PMID: 11481332	MDCK II	2001
3	PMID: 12470645	HeLa	2002
4	PMID: 11944901	Yeast	2002
5	PMID: 12376546	HeLa	2002
6	PMID: 12857874	HeLa	2003
7	PMID: 15304524	HeLa; A431	2004
8	PMID: 16148947	HeLa; KE37; CHO	2005
9	PMID: 15601896	HeLa; NIH-3T3	2005
10	PMID: 17007872	HeLa	2006
11	PMID: 17030804	HeLa	2006
12	PMID: 17628206	HeLa	2007
13	PMID: 17229837	U2OS; 293T	2007
14	PMID: 18511905	HeLa	2008
15	PMID: 18511905	HeLa	2008
16	PMID: 19761540	HFF; HK293	2009
17	PMID: 20682791	HEK293; MEF; HeLa	2010
18	PMID: 20727405	HEK293	2011
19	PMID: 25035494	RPE1	2014

Figure 2. The expression pattern of Rab11fip3 (A) and Rab11fip4 (B) protein across the human body. Data retrieved from the Human Proteome Map database (<https://www.humanproteomemap.org>).

Figure 3. The expression pattern of Rab11fip3 (A) and Rab11fip4 (B) protein across the human body. Data retrieved from the Human Protein Atlas database (<https://www.proteinatlas.org/>).

-All the graphs show that uninfected and untreated cells have a wide dispersion of the mean values calculated. 10 cells were analyzed for this condition. However, for all other treatments the number of the cells analyzed was lower, varying from 4-6 cells. The case of Latrunculin A treatment for example of Fig 2, the cells have a bimodal behavior, with some being affected by the treatment and others not. Analyzing the same number of cells is preferential, especially considering the dispersion of untreated samples.

We analyzed thousands of tracks per cell, which were distilled down to one value, so although the result may appear as if it is derived from only a few data points, these actually represent many thousands of tracks. We find the data to be very robust and consistent between cells.

Solid conclusions should be derived from truly independent samples. Despite counting thousands of tracks the authors are in reality analysing the behaviour of 4 immortalized cells (carrying many mutations), which is by no means representative of the behaviour of cells or of viruses (carrying many mutations). The error associated with the data of Fig 3-6 supports this view.

>**Author Response August 2019:** *In the revised text we have increased the number of cells so that all three main conclusions are based on >10,000 tracks from 7-10 cells per condition. This additional analysis did not change any of the conclusions in the manuscript.*

- The number of moving Rab11/RNPs is very different between conditions. Bigger aggregates are brighter than the smaller versions, which will bias the analyses and acquisition of moving targets. This is particular clear when analyzing S6-S8. Please comment. The inclusion of the number of moving vesicles would complement the data. Also, actin/microtubule-based movements have very different behaviors, and different cellular localization. The methodology described would allow discriminating movements relatively to their cellular location -close to the nucleus or to the plasma membrane, which would be an insightful way to look at Rab11 positioning during viral infection, showing novel and relevant alterations to this molecule for IA, IB and RSV.

Data in the original S6 and S8 figures were tracking influenza vRNP not Rab11A-GFP. In our system we have not observed ' aggregates'

of PA-GFP or Rab11A-GFP in live cells. Also our tracking algorithms include all spot intensities in the tracking analysis. Supplemental figure S15 displays violin plots of the Rab11A foci intensity in tracks from all cells of various conditions. These data conclusively demonstrate similar signal intensities in Rab11A untreated, nocodazole treated, latrunculin treated, uninfected and infected cells. Therefore, we do not believe that there is a bias in the tracking based on spot intensities. We do agree with the Reviewer that examination of Rab11A movement in relation to cellular features would be powerful. However, three-dimensional intracellular tracking of spots is a difficult problem and we are working closely with cell biologists to develop additional algorithms to answer these questions. At this time, we feel that they are beyond the scope of this study.

This reviewer is not convinced by the violin plots on the similar Rab11 foci intensities for all conditions: the median, higher values and probability differ between treatments. Changes in Rab11 intensity foci have been reported independently by the labs of Kawaoka and Digard and quantitatively assessed in Vale-Costa et al, 2016.

However, as this issue does not affect the main conclusions of the manuscript, and the authors are unable to examine Rab11/RNP movements in relation to cellular structures, this point becomes irrelevant.

>**Author Response August 2019:** *We agree this point does not impact the main conclusions of our manuscript.*

ADDITIONAL COMMENTS:

- There are many articles on Myo5B binding to Rab11 via FIP2. To strengthen the data, the authors should validate the uncorrelated behaviour of Myo5B/FIP2 during infection and comment on it.

- In the absence of infection, what is the effect of reducing Rab11-dynein interaction on Rab11 vesicular movement? Also, does the overexpression of dynein interfere with viral infection or with Rab11 movement? As a premise of the direct binding between dynein and Rab11, overexpressing dynein should interfere with Rab11 arrest observed during infection.

>**Author Response August 2019:** *Reduction of total dynein levels will impact all vesicular transport in the cell, thus this suggestion will not provide clarity to the mechanism relevant to influenza viral infection. We have added a schematic (figure 8) that represents our working model.*

- Page 15, lines 30-33

“ Recent work using mitochondrial-targeting of FIP proteins suggested that IAV vRNP segments outcompete FIP2 for binding with Rab11A 45. Consistent with this finding we observed a significant decrease in the association with FIP2 binding to Rab11A during IAV infection (Figure 7b). However, the levels of all other FIPs were not altered” .

Please refer to the work of Momose et al, Plos One, 2011, the first manuscript to analyse FIPs in influenza A virus infected cells. Also, the final sentence needs revision. The mass spectrometry data of both the IP and cell input did not detect FIP3 or FIP4. Therefore, it is inaccurate to state that all other FIPs were not altered because the set of FIPs is incomplete.

> **Author Response August 2019:** *We have modified the text as stated (page 8 line 44 and page 9 lines 2-6).*

Discussion:

- The reduction in dynein is interesting. It has implications to the mechanisms reported on the interactions between molecular motors and Rab11 vesicles, and how the tug of war between different motors contribute to Rab11 movement. The authors should discuss this. In addition, how would a specific reduction in dynein affect Rab11 movement in polarized cells?

- Page 16 Lines 29-36

> **Author Response August 2019:** *This point is now discussed on lines page 10 lines 22-25.*

The work of Momose et al, Plos One, 2011, and the final sentence needs revision for the reasons explained above.

- Page 16 Lines 36-38

> **Author Response August 2019:** *We have modified the text as stated (page 8 line 44 and page 9 lines 2-6).*

Please revise your sentence after reanalysing the manuscripts, as you find suitable. What I read from the papers cited is:

The molecular motors proposed to be reduced during influenza infection are the ones operating via FIPs, like dynein (in article 45), but KIF13 does not belong to this group as it binds Rab11 independently of FIPs (in article 46).

> **Author Response August 2019:** *We disagree. Rab11A IP/Mass Spectrometry did not identify KIF13 as a interactor of Rab11A in A549 cells and our included IP did*

not bring down KIF13. 'Article 46' (PMID:29061883) also does not demonstrate direct interaction of Rab11A and KIF13. In contrast we clearly show that association with dynein is reduced during viral infection.

- Page 16, Line 42-44

for a microtubule independent mechanism of IAV vRNP transport that may be Rab11A-independent as well.

Please include the paper of Avilov et al, JVI, 2012 and comment on their identification that actin is important for vRNP transport.

Also, the authors use a system overexpressing GFP-Rab11, that although carefully evaluated, is also not the ideal, especially if maintaining endogenous Rab11 in the cell. This means that the percentage of RNPs that bind GFP-Rab11 is underestimated and that the conclusion on “ a microtubule independent mechanism of IAV vRNP transport that may be Rab11A-independent as well” is overestimated. Please comment.

> **Author Response August 2019:** *We have included the relevant citation.*

- Page 17, Line 18 - 22:

“ However, recent work has suggested that vRNP exit the ER in liquidphase organelles rather than membranous structures⁵¹. Therefore, additional work examining the intracellular movement of individual vRNP segments and ER membranes within a single cell is needed to elucidate the importance of dynamic membranous scaffolds during IAV infection” .

This comment should be revised. The paper Milovanovic, D. et al, 2018, Science (10.1126/science.aat5671) (Pietro di Camilli Lab) shows that vesicles destined for the immunological synapse are concentrated in liquid organelles. Similarly, the liquid organelles described in article 51 contain vesicles in the core and are not mutually exclusive to membranous structures assisting genome assembly.

> **Author Response August 2019:** *This sentence has been modified, see page 11 line 6.*

REVIEWERS' COMMENTS:

Reviewer #1 (Remarks to the Author):

The reviewing task was complicated by the fact that the page/line numbering in the point by point response letter was not matched to the page/line numbering of the revised manuscript, and all changes were not highlighted.

However my analysis is that the major concerns raised at the previous round of revision have been satisfactorily addressed by the authors.

Please note that the legend for Figure 8 is missing.